



# A one-dimensional urban flow model with an Eddy-diffusivity Mass-flux (EDMF) scheme and refined turbulent transport (MLUCM v3.0)

Jiachen Lu[1,2], Negin Nazarian[1,2], Melissa Anne Hart[2], E. Scott Krayenhoff[3], and Alberto Martilli[4]

[1]School of Built Environment, University of New South Wales, Sydney, Australia
[2]ARC Centre of Excellence for Climate Extremes, University of New South Wales, Sydney, Australia
[3]School of Environmental Sciences, University of Guelph, Guelph, ON, Canada
[4]Atmospheric Pollution Division, Environmental Department, CIEMAT, Madrid, Spain

**Correspondence:** Jiachen Lu (jiachen.lu@unsw.edu.au)

**Abstract.**

In recent years, urban canopy models (UCMs) have been used as fully coupled components of mesoscale atmospheric models as well as offline tools to estimate temperature and surface fluxes using atmospheric forcings. Examples include multi-layer urban canopy models (MLUCMs), where the vertical variability of turbulent fluxes is calculated by solving prognostic momen-

tum and turbulent kinetic energy (TKE, $k$) equations using length scale ($l$) and drag parameterizations. These parameterizations are based on the well-established 1.5-order $k-l$ turbulence closure theory and are often informed by microscale fluid dynamics simulations. However, this approach can include simplifications such as the assumption of the same diffusion coefficient for momentum, TKE, and scalars. In addition, the dispersive stresses arising from spatially-averaged flow properties have been parameterized together with the turbulent fluxes while being controlled by different mechanisms. Both of these assumptions

impact the quantification of turbulent exchange of flow properties and subsequent air temperature prediction in urban canopies. To assess these assumptions and improve corresponding parameterization, we conducted 49 large-eddy simulations (LES) for idealized urban arrays, encompassing variable building height distributions and a comprehensive range of urban densities ($\lambda_p \in [0.0625, 0.64]$) seen in global cities. We find that the efficiency of turbulent transport (numerically described via diffusion coefficients) is similar for scalars and momentum but 3.5 times higher for TKE. Additionally, the parameterization of the

dispersive momentum flux using the $k-l$ closure was a source of error, while scaling with the pressure gradient and urban morphological parameters appears more appropriate. In response to these findings, we propose two changes to MLUCM v2.0: (a) separate characterization for turbulent diffusion coefficient for momentum and TKE; and (b) introduction of an explicit physics-based "mass flux" term to represent the non-Gaussian component of the dispersive momentum transport as an amendment to the existing "eddy diffusivity" framework. The updated one-dimensional model, after being tuned for building height

variability, is further compared against the original LES results and demonstrates improved performance in predicting vertical turbulent exchange in urban canopies.



## 1   Introduction

The urban canopy is a unique and complex land cover type in climate models (Oke et al., 2017). As climate models become more capable of high-resolution simulations, the challenge resides in developing and validating urban canopy models (UCMs)
to account for finer-scale interactions with complex urban forms. Applications such as running UCMs to accurately describe urban surface fluxes to atmospheric boundary layers (Schoetter et al., 2020) and UCMs as offline models coupling with mesoscale forcing to simulate in-canopy processes (Redon et al., 2020) can then benefit from such infrastructure developments.

The development of UCMs relies on the similarity between flow through the urban canopy layer (UCL) and other more commonly investigated types of flow. For example, Macdonald (2000) modified the exponential wind profile from flow over
vegetative canopy (Finnigan, 2000) based on the aerodynamic roughness properties of the urban surface to predict the urban wind speed that is commonly used in single-layer urban canopy models (SLUCM, (Kusaka et al., 2001)). The 1.5-order turbulence closure scheme (Bougeault and Lacarrere, 1989) used for turbulence parameterization in the Planetary Boundary Layer (PBL) was also extended to the UCL, facilitating the development of the multi-layer urban canopy model (MLUCM, (Santiago and Martilli, 2010a)). The urban form and fabric, however, feature distinct behaviors that require modifications to length scales
(Cheng and Porté-Agel, 2021) and form drag (Sützl et al., 2021b) that are uniquely developed to relate urban morphological parameters (Lu et al., 2023b). In addition, some physical processes commonly seen in urban environments, such as thermal instability, also result in unique flow characteristics in urban canopies that have been separately evaluated in urban canopy models (Santiago et al., 2014; Simón-moral et al., 2016; Nazarian and Kleissl, 2015).

The modeling of urban surfaces develops along with the capability of computational power and availability of realistic
building morphology (Sirko et al., 2021). As the community is pushing the resolution of regional climate models to sub-kilometer (Sützl et al., 2021a), there is an urgent need for a more detailed characterization of finer(subgrid)-scale (and higher-order) urban phenomenon (such as building wake and canyon re-circulation (Oke et al., 2017)). The most common way to enable UCMs to capture these factors is by relating the variation of urban geometric parameters to the corresponding flow statistics based on Computational Fluid Dynamics (CFD) models such as Large-Eddy Simulation (LES) that optimally balance
accuracy and cost (Blocken, 2018; Nazarian et al., 2020).

The development of UCMs includes designing modules to predict flow properties such as wind speed (Castro, 2017), TKE (Christen et al., 2009), and scalars (Lim et al., 2022) that interact with urban structures differently. For example, increasing the frontal density of roughness elements was found to amplify the turbulent momentum transport but hinder turbulent scalar transport (Li and Bou-Zeid, 2019). However, efforts to identify and characterize the dissimilarity of the transport mechanisms
of flow properties are rare due to the lack of experimental evidence to support quantitative characterizations. Therefore, momentum, TKE, and scalar diffusion efficiency in modeling practice are typically assumed to be the same (Martilli et al., 2015; Nazarian et al., 2020), introducing errors that may degrade model performance.

Simulations with MLUCMs are particularly sensitive to details of the parameterization that influence the strength of turbulence intensity within the canopy that facilitates turbulent transports of flow properties. Thus, accurate quantification of TKE
is essential; however, previous work leading to the development of MLUCM v2.0 has shown an underestimation of canopy



TKE (Nazarian et al., 2020). Moreover, canopy TKE must be accurately quantified for simplified neighborhoods before the addition of the flow effects arising from the added complexity of natural urban areas, such as sub-facet scale structures (overhangs, awnings, irregular building shapes, etc.) or trees (Krayenhoff et al., 2020). Indeed, a vital advantage of the multi-layer representation of urban canopies relative to single-layer models is the ability to vertically resolve the drag and TKE production
elements (buildings, trees, etc.) and thus more accurately capture the profiles of wind and TKE and associated implications for convection and ventilation.

Arising from spatially-averaged flow properties (that is core to MLUCM development), dispersive fluxes illustrate the transport of variables by time mean structures smaller than the averaging grid size, constituting another unique urban phenomenon (Poggi and Katul, 2008b). These dispersive structures strongly depend on the complexity of the underlying urban geometry (Lu
et al., 2023b) that forbids generalized characterizations. Therefore, dispersive flux has only been pragmatically considered in (Nazarian et al., 2020; Simón-moral et al., 2016) based on the K-theory framework as an increment to its turbulent counterpart. In practice, this approach resulted in an enhanced eddy diffusivity and implicitly assumed the correlation between dispersive flux and turbulence intensity and the vertical gradient of the flow property. However, the physical basis of such a correlation is still unclear and may not comprehensively improve the predictability of UCMs. Instead, the relative strength of the dispersive
flux is highly related to the underlying canopy geometry and exhibits substantial spatial variability (Harman et al., 2016). Varying with the shape of roughness elements, the dispersive component can be about $50\%$ of the total transport over non-cubic blocks (Li and Bou-Zeid, 2019). Further, realistic urban neighborhoods induce even higher dispersive contributions due to the more substantial spatial flow variability (Akinlabi et al., 2022; Giometto et al., 2016). Consequently, the lack of dispersive stress characterization restricts the study of canopy flows to relatively less complex and homogeneous flow, such as on gentle
topography (e.g., (Finnigan and Belcher, 2004)) with slowly varying packing density (Ross, 2012).

Based on the above fundamental deficiencies embedded in the multi-layer model, better performance of UCMs on more complex flows requires finer and more physically-based characterization of dispersive fluxes. In this research, based on a recently developed urban flow dataset (Lu et al., 2023a), we aim to assess and improve the prediction of the TKE and momentum in the urban canopy through a more accurate parameterization of turbulent transport in the MLUCM (Nazarian et al., 2020) in
two ways:

- – Separate characterization of transport efficiency of different flow properties.

- – Introduce a physics-based modeling of the dispersive transport of momentum.

This paper first describes the flow dataset using the PArallelized Large-Eddy Simulation Model (PALM) in Sect. 2.1. Then different components of MLUCM v2.0 (Nazarian et al., 2020) are revisited for the conventional eddy diffusion module in Sect.
3.1 and for the drag and dissipation module in Sect. 3.3. A novel physically-based mass-flux parameterization is proposed based on the pressure gradient scaling and the geometric condition of the neighborhood in Sect. 3.2. In Sect. 4, the proposed changes for MLUCM are tested and compared across 49 urban arrays with different horizontal (staggered, aligned) and vertical (height variabilities) arrangements. Section 5 summarizes the findings of this study and provides perspectives for future developments of the multi-layer model.



## 2 Methodology

In this study, we used two methods to assess the turbulent transport behavior of the urban canopy flow, i.e., computational fluid dynamics (CFD) simulations and an offline multi-layer urban canopy model. In Sect. 2.1, with the help of LES simulations recording the third-order moments in turbulent flows, we evaluate differences in turbulent transport of momentum, TKE, and scalar. Following the differences, we further revisit the turbulent parameterization in Sect. 3.1 and Sect. 3.2 followed by a comprehensive evaluation of model performance against LES in Sect. 4.

### 2.1 Large-Eddy Simulation (LES) setup and dataset

Conventionally, two configurations of idealized building arrays, "aligned" and "staggered" (Coceal et al., 2006), are employed to approximate horizontal arrangements of real neighborhoods. The aligned array simulates the situation in natural communities where streets align with the prevailing wind direction. The staggered array, in contrast, is mainly employed to calibrate UCM (Santiago and Martilli, 2010b; McNorton et al., 2021) in that it has no significant corridors (i.e., streets) aligned with the wind, potentially resembling the frequent situation where wind and street directions do not align. Both choices simplify realistic urban neighborhoods, but staggered arrays provide a closer approximation to average conditions in real cities (Nazarian et al., 2020). Accordingly, we select the staggered configuration to conduct LES simulations of urban wind flow to determine parameterizations in MLUCM.

Simulations are conducted in the Parallelized Large-eddy Simulation Model (PALM, version r4554) (Maronga et al., 2020) with $\lambda_p$ ranging from 0.0625 to 0.64, representing urban densities seen in global cities. The numerical setup follows (Lu et al., 2023a; Nazarian et al., 2020; Lu et al., 2023b), which has been validated against direct numerical simulation (DNS) (Coceal et al., 2007) and wind tunnel experiments (Brown et al., 2001) but will be listed here for completeness.

The computational domain is discretized equidistantly using second-order central differences (Piacsek and Williams, 1970) horizontal and staggered Arakawa C-grid in the vertical direction. The momentum field is solved using the filtered prognostic incompressible Boussinesq equations with the time integration following a minimal storage scheme (Williamson, 1980). The covariance terms from the filtering procedure were parameterized using a 1.5-order closure scheme after (Deardorff, 1980). The pressure perturbation was calculated in Poisson's equation and was solved by the FFTW scheme (Frigo and Johnson, 1998). Apart from solving momentum and pressure fields, a passive scalar was introduced with a surface value $c_0$=0.06, and a constant negative gradient of -5e-5/m forms the initial profile. At the same time, the total scalar concentration was maintained by two Neumann boundary conditions with surface emission equal to 0.0001 and a sink at the domain top equal to 0.0001*(1-$\lambda_{p,0}$) where $\lambda_{p,0}$ indicate urban density at the surface.

The detailed design of building arrays can be found in (Lu et al., 2023a) and is briefly explained here: While maintaining the same volume of roughness elements (i.e., same mean building height, $H_{mean}$=16m), we consider three height distributions, i.e., $H_{std} \in [0m, 2.8m, 5.6m]$ which result in different maximum building height $H_{max}$. Additionally, we consider three types of urban arrays: a) **Continuous** distribution of building height with a relatively low $H_{max}$; b) **Clustered** configuration with two clusters (three buildings) of low and tall building blocks; and c) **High-rise** configuration, featuring one



**Table 1.** Dataset details for 49 urban arrays discussed in this study. Four building height arrangements (uniform, Continuous, Clustered, and High-rise) with two standard deviations of height configurations are considered. The maximum $H_{\max}$ and minimum $H_{\min}$ building heights and a 3D view of example ($\lambda_p = 0.25$) are shown below each category.

| *Configuration* | Uniform | Continuous | | Clustered | | High-rise | |
|---|---|---|---|---|---|---|---|
| ***Staggered*** | | | | | | | |
| Hstd [m] | 0 | 2.8 | 5.6 | 2.8 | 5.6 | 2.8 | 5.6 |
| Hmean [m] | 16 | 16 | 16 | 16 | 16 | 16 | 16 |
| Hmin [m] | 16 | 10 | 4 | 11 | 6 | 12.5 | 9 |
| Hmax [m] | 16 | 20 | 24 | 21 | 26 | 26.5 | 37 |
| Topography Ex ($\lambda_p$=0.25, $H_{\mathrm{std}}$=5.6m) | | | | | | | |

tall and a cluster (three) of low building blocks. Each configuration is then run with seven urban packing densities, i.e., $\lambda_p \in [0.0625, 0.1111, 0.16, 0.25, 0.35, 0.4444, 0.64]$, which yields a total of 49 simulations cases. Details of configurations considered are shown in Table 1.

## 3   Assessment of the Multi-layer Urban Canyon Model (MLUCM)

In mesoscale climate models, flow through canopies is often simulated at scales much larger than the typical surface processes, such as turbulent eddies and obstacle wakes. A common approach in this situation is to apply a time-averaging over an interval longer than the scale of the slowest eddies and a spatial-averaging over a length larger than the typical spatial deviations in the flow (Raupach and Shaw, 1982; Finnigan, 1985). For momentum, Reynolds decomposition is firstly applied to the 3-dimensional instantaneous equations that decompose mean flow quantities from their fluctuating components (time or ensemble averaging, $\phi = \overline{\phi} + \phi'$). Then, flow properties are spatially averaged to match a grid cell of a mesoscale model (horizontal averaging, $\overline{\phi} = \langle \overline{\phi} \rangle + \tilde{\phi}$). In urban canopies where the volume fraction occupied by obstacles is not negligible, the intrinsic average (Schmid et al., 2019) that excludes the roughness fraction in the spatially-averaging is more favorable given that additional terms compensating for the volume fraction occupied by roughness elements in the canopy are included (Blunn et al., 2022). The resulting one-dimensional representation of the momentum and passive scalar equation assuming horizontal homogeneity, negligible Coriolis, and buoyancy force reads,

$$\frac{\partial \langle \bar{u} \rangle}{\partial t} = -\frac{\partial \left( \langle \overline{u'w'} \rangle + \langle \tilde{u}\tilde{w} \rangle \right)}{\partial z} - \frac{\langle \overline{u'w'} \rangle + \langle \tilde{u}\tilde{w} \rangle}{\gamma}\frac{\partial \gamma}{\partial z} + f_i - \frac{1}{\rho}\left\langle \frac{\partial P}{\partial x_i} \right\rangle + \nu \left\langle \nabla^2 \tilde{u} \right\rangle \tag{1}$$

$$\frac{\partial \langle \bar{c} \rangle}{\partial t} = -\frac{\partial \left( \langle \overline{c'w'} \rangle + \langle \tilde{c}\tilde{w} \rangle \right)}{\partial z} - \frac{\langle \overline{c'w'} \rangle + \langle \tilde{c}\tilde{w} \rangle}{\gamma}\frac{\partial \gamma}{\partial z} + c_i \tag{2}$$



where $\rho$ is the air density, $u$ and $w$ is the streamwise and vertical velocity so that $\langle \overline{u'w'} \rangle$ and $\langle \tilde{u}\tilde{w} \rangle$ forms the spatially-averaged turbulent and dispersive Reynolds stress respectively. The first term on the RHS of both equations represents transport events, while the second term is a term that compensates for volume change in the intrinsic averaging, where $\gamma(z) = 1 - \lambda_p(z)$ is the fluid fraction at the vertical index $z$. The third term, $f_i$ and $c_i$, account for external sources or sinks, such as pressure gradient driving the flow and surface emission of pollutants. The fourth term of Eq. 1 represents a term risen from spatially averaging that accounts for momentum sink due to form and skin drag. The multi-layer model is then designed to solve these equations to parameterize the effects of turbulence, drags, and vertical transport of momentum and scalars.

### 3.1 Characterization of the eddy diffusivity for different flow statistics

The Reynolds averaging over momentum equation yields nonlinear terms that must be parameterized to close the equation. The most common approach in parameterizing the Reynolds stress in UCMs is based on the $K$ theory (as also seen in models compared in (Best and Grimmond, 2015; Lipson et al., 2023)) to represent the efficiency of turbulent transports by employing the eddy diffusivity coefficient, together with the vertical gradient of the flow properties (For momentum, $\langle K_m \rangle = \langle \overline{u'w'} \rangle / \langle \frac{\partial \overline{u}}{\partial z} \rangle$). In a variety of ways specifying the eddy diffusivity, the 1.5-order closure (Martilli et al., 2002) assuming the transport strength relating to the turbulence intensity is commonly employed for urban canopy flow,

$$K_m = C_k l_k \langle k \rangle^{1/2} \tag{3}$$

where $C_k$ is a model constant, $l_k$ is a turbulent length scale, and $k$ is TKE that has to be prognostically obtained during the simulation follows Eq. 4.

$$\frac{\partial \langle \bar{k} \rangle}{\partial t} = -\frac{\partial \left( \langle \overline{k'w'} \rangle + \langle \tilde{k}\tilde{w} \rangle \right)}{\partial z} - \frac{\langle \overline{k'w'} \rangle + \langle \tilde{k}\tilde{w} \rangle}{\gamma} \frac{\partial \gamma}{\partial z} - \left\langle \overline{u_i'u_j'} \right\rangle \frac{\partial \langle \overline{u_i} \rangle}{\partial x_j} - \left\langle \widetilde{u_i'u_j'} \frac{\partial \tilde{u}_i}{\partial x_j} \right\rangle - D_k - \langle \bar{\varepsilon} \rangle \tag{4}$$

Similar to the momentum equation Eq 1, the first two terms on RHS represent the transport of TKE, followed by two terms for shear and dispersive TKE production, respectively. The fourth term represents the source of TKE generated through the interaction with the buildings and the airflow, while the last term $\langle \bar{\varepsilon} \rangle$ is the viscous dissipation of TKE that can be parameterized as,

$$\langle \bar{\varepsilon} \rangle = C_\varepsilon \frac{\langle \bar{k} \rangle^{3/2}}{l_\varepsilon} \tag{5}$$

where $C_\varepsilon$ and $l_\varepsilon$ are the model constant and the dissipation length scale, respectively, and have a relationship with turbulent length scale based on the turbulent viscosity $C_\mu$ (Santiago and Martilli, 2010b) that measures the relative strength of viscous dissipation over turbulent transport,

$$C_k l_k = C_\mu \frac{l_\varepsilon}{C_\varepsilon} \tag{6}$$



To date, most models assume the same value for the diffusion coefficient in the momentum, TKE, and scalar equations, possibly due to the lack of data on TKE and scalar budget terms (note that the explicit evaluation of TKE transport and diffusivity, in particular, is only possible from numerical simulations that resolve third-order moments, e.g., LES and direct numerical simulations (DNS)). However, previous LES analyses of urban flow indicated a different transport efficiency of varying flow properties (Nazarian et al., 2020). Accordingly, we distinguish between three diffusion coefficients as,

$$\langle K_m \rangle = \langle \overline{u'w'} \rangle / \left\langle \frac{\partial \overline{u}}{\partial z} \right\rangle = \langle \overline{u'w'} \rangle / \left( \frac{\partial \langle \overline{u} \rangle}{\partial z} + \frac{\langle \overline{u} \rangle}{\gamma} \frac{\partial \gamma}{\partial z} \right) \tag{7a}$$

$$\langle K_c \rangle = \langle \overline{c'w'} \rangle / \left\langle \frac{\partial \overline{c}}{\partial z} \right\rangle = \langle \overline{c'w'} \rangle / \left( \frac{\partial \langle \overline{c} \rangle}{\partial z} + \frac{\langle \overline{c} \rangle}{\gamma} \frac{\partial \gamma}{\partial z} \right) \tag{7b}$$

$$\langle K_k \rangle = \langle \overline{k'w'} \rangle / \left\langle \frac{\partial \overline{k}}{\partial z} \right\rangle = \langle \overline{k'w'} \rangle / \left( \frac{\partial \langle \overline{k} \rangle}{\partial z} + \frac{\langle \overline{k} \rangle}{\gamma} \frac{\partial \gamma}{\partial z} \right) \tag{7c}$$

The turbulent fluxes, vertical gradient, and eddy diffusivity of momentum, TKE, and scalar are shown in Fig. 1 for staggered-uniform configurations. Note that the profiles over other designs with building height variability (not shown here) share similar characteristics. Therefore, we chose the uniform-height layout to reveal the difference among flow properties better.

Within the canopy, the turbulent fluxes of momentum and TKE are overall negative due to the resistance difference to the constant pressure gradient between the free atmosphere and urban canopy. The turbulent fluxes of passive scalars are positive due to the surface emission being the only source and exhibiting a more minor variation across different densities. Being first flow moments, the eddy diffusivity for scalar ($K_c$) and momentum ($K_m$) maintains a similar shape as a result of the relatively simpler mechanism in their production, destruction, and transport which justifies the simplification ($K_c \sim K_m$) that is consistent with Li and Bou-Zeid (2019).

However, the eddy diffusivity of TKE ($K_k$) demonstrates a greater complexity with some negative values at the ground and canopy interface that marks the counter-gradient transport of TKE indicating two vital local shear production regions: a) The canopy-top shear zone is relatively well-studied that induced by clearings of roughness elements Giometto et al. (2016); b) near-ground shear zone that is unique in medium-dense layouts ($\lambda_p \leq 0.25$) where flow reversal appears and counteracts with the pressure gradient, inducing significant shear source of TKE Lu et al. (2023a). This lack of consideration of the near-ground flow features partially explains the underestimation of local TKE seen in (Nazarian et al., 2020) and can be mitigated by imposing a height-dependent model constant (e.g., (Glazunov et al., 2021)), but is beyond the scope of this study that aims to refine the turbulent transport characterization within the whole canopy. Apart from the near-ground and canopy-top inversion, $K_k$ shares a similar shape with $K_m$ but presents a significantly larger magnitude, which indicates models using unified $K$ underestimate the transport of TKE and require closer characterization.

The distinct behavior of the densest case (purple lines in Fig. 1-c) comes from the positive gradient of streamwise velocity in the middle of the canopy and positive Reynolds stress in the lower half of the canopy that is hard to characterize. Such a high density is not rare, especially in some European and South American cities, generally with different horizontal arrangements, resulting in very different flow characteristics (Lu et al., 2023b) that do not break the parameterization in the present study.





**Figure 1.** Vertical profiles of turbulent flux (1st column), vertical gradient (2nd column), and eddy diffusivity (3rd column) of momentum, TKE, and scalar flux from Eq. 7 evaluated over uniform-staggered configurations. The yellow-colored range indicates the mean building height. The x-axis is limited to show most profiles for better presentability. Values are off the axis limit at the canopy top due to the strong shear zone.





Following the parameterization of length scales in (Nazarian et al., 2020), we average the profile of eddy-diffusivities shown in Fig. 1 up to $H_{\mathrm{mean}}$ to reveal the integral behavior of transport behaviors across configurations. Figure 2 presents the scatter

of the vertically averaged eddy-diffusivity for momentum $K_{m,T}$ and TKE $K_{k,T}$ without considering dispersive flux, and those include the dispersive contribution ($K_{m,T+D}$ and $K_{k,T+D}$). Characterization of dispersive fluxes within the K-theory framework enhances the exchange rate of flow properties, which agrees with previous studies such as (Akinlabi et al., 2022) where the dispersive flux is mostly downwards.

The magnitude of eddy-diffusivity for TKE is $\sim 3.5$ times higher than that for momentum (note the vertical range of two rows

in Fig. 1). This considerable difference demonstrates a significant contrast to the parameterization strategy in (Nazarian et al., 2020) assumed $K_{m,T+D} = K_{k,T+D}$ that results in underestimations of TKE within the canopy. In addition, unlike $K_{m,T+D}$, which generally decreases with urban density before the layout becomes very dense, $K_{k,T+D}$ exhibits a less decisive trend to density. The magnitude difference and variations over densities between $K_{m,T+D}$ and $K_{k,T+D}$ urges a distinction in their parameterizations.

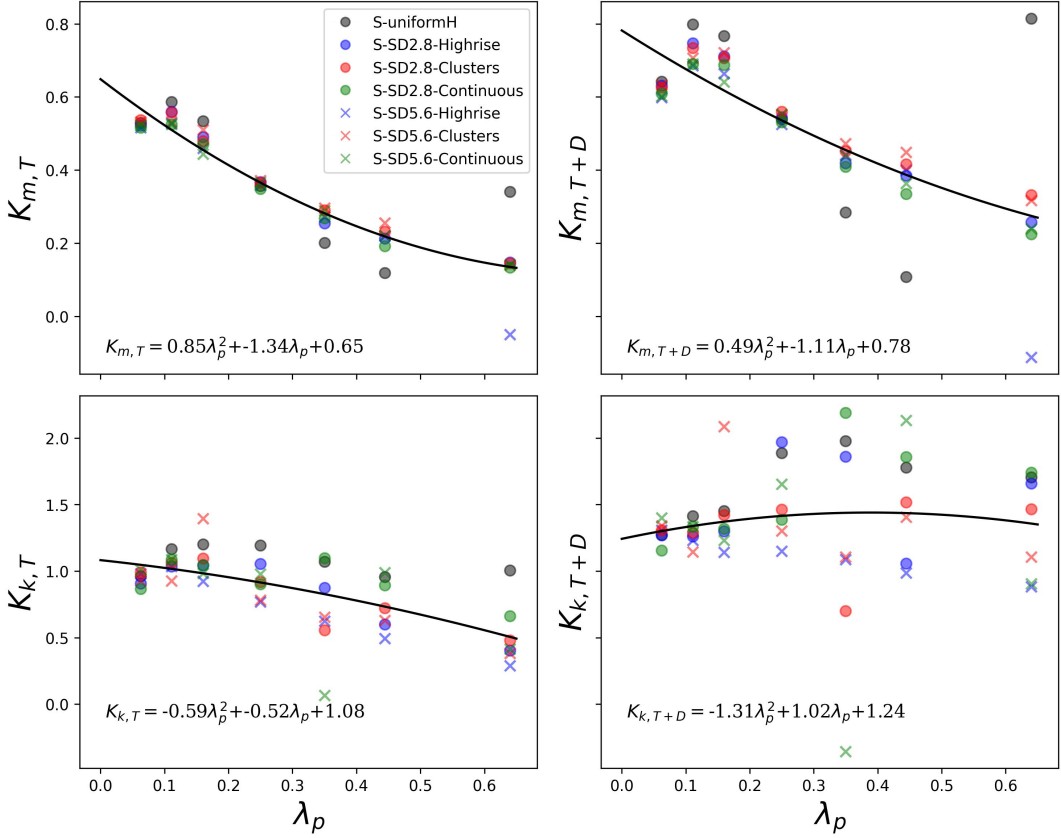

**Figure 2.** Canopy-averaged eddy-diffusivity for momentum and TKE with ($K_{m,T+D}$ and $K_{k,T+D}$) and without ($K_{m,T}$ and $K_{k,T}$) additional consideration of the dispersive component. Black lines indicate second-order polynomial regressions over means of seven configurations under the same densities where the regression equation is shown at the bottom of each subplot.





The eddy diffusivity was not directly parameterized under the 1.5-order closure framework developed in Martilli et al. (2002) but as a turbulent length scale correlated with the dissipation length scale in Eq. 6 (i.e., $C_k l_k = C_\mu \frac{l_\varepsilon}{C_\varepsilon}$). Here we follow the same approach in Nazarian et al. (2020) to parameterize the turbulent viscosity ($C_\mu$) and dissipation length scale ($\frac{l_\varepsilon}{C_\varepsilon}$) to obtain the turbulent length scale ($C_k l_k$). Considering the parameterization for dissipation length scale is the same across scenarios considered in Table 2, the difference in the transport efficiency demonstrated in Sect. 3.1 and Sect. 3.2 can be expressed as the

difference in the turbulent viscosity ($C_\mu$) as shown in Fig. 3.

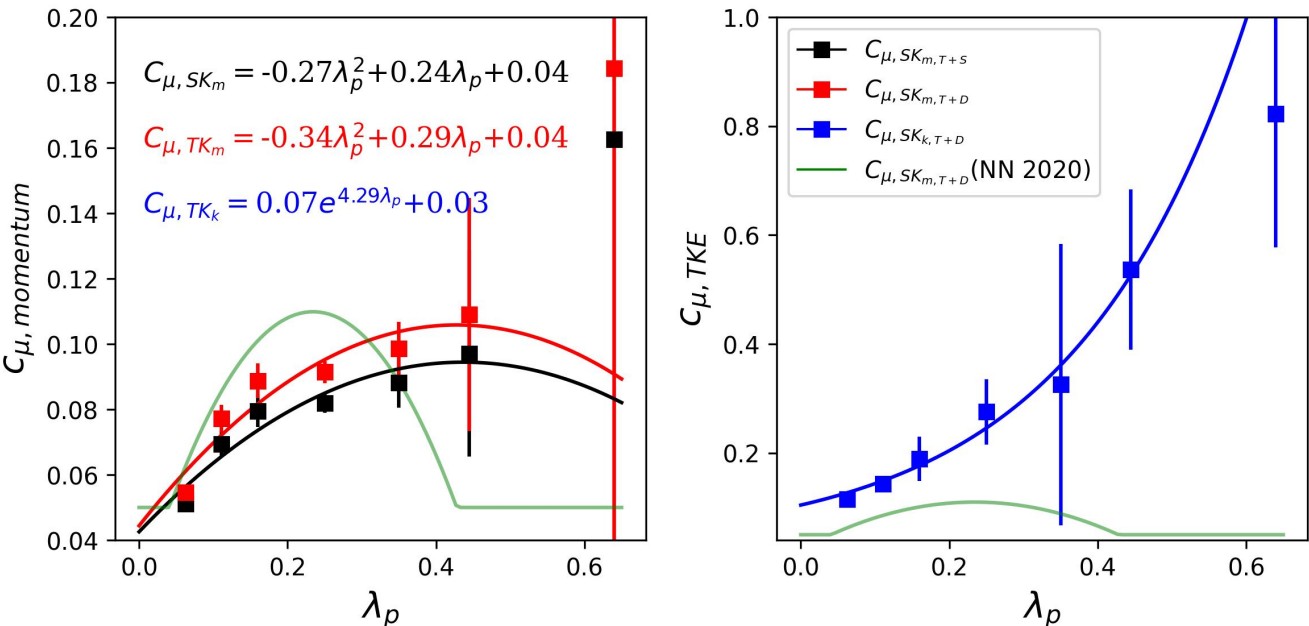

**Figure 3.** Canopy-averaged turbulent viscosity for momentum and TKE. The left panel shows the turbulent viscosity of momentum where the red-colored and black-colored dots indicate evaluation adding the dispersive with turbulent fluxes ($C_{\mu,TK_m}$) and sampling out the non-Gaussian component of the dispersive flux ($C_{\mu,SK_m}$) and lumping the residue, respectively. The right panel shows the turbulent viscosity of TKE, lumping the dispersive component. Data points are binned by their ground density $\lambda_p(z=0)$, where the horizontal line attached to each square indicates the standard deviation among the data points binned. Green-colored lines indicate the parameterization from Nazarian et al. (2020).

   Figure 3 shows the calculated turbulent viscosity in 49 idealized urban arrays binned over densities for better presentation, where the original parameterization with seven uniform-height cases (Nazarian et al., 2020) is shown in green lines. Generally, $C_\mu$ increases with density over sparse and medium layouts, which indicates the overweighting trend of viscous dissipation of of TKE over the turbulent transport that is consistent with the transition of the flow regime (Oke et al., 2017). Meanwhile,

denser layouts amplify the impact of building height variability, leading to higher standard deviation among binned scenarios, which is consistent with Lu et al. (2023a). Turbulent viscosity for momentum lumping the dispersive flux $C_{\mu,K_m,T+D}$ shows a good agreement with that from (Nazarian et al., 2020) except for the peaking at higher density. Due to separate explicit



characterization of the non-Gaussian dispersive momentum transport, $C_{\mu,K_{m,T+S}}$ is slightly lower than $C_{\mu,K_{m,T+D}}$ but shows a similar shape. The turbulent viscosity of TKE ($C_{\mu,K_{k,T+D}}$) is about 3.5 times higher than that of momentum (note the difference on the y-axis limit) and shows exponential growth over density.

## 3.2 Characterization of the dispersive momentum transport

The momentum transport in urban canopy flow is also modulated by its spatial variability (Eq. 1) induced by rigid volume of roughness elements that are hard to characterize due to its extreme heterogeneity (Poggi and Katul, 2008b). Nevertheless, flow heterogeneity is not unique to the urban boundary layer but is prevalent in the real-world atmosphere layer due to the surface heating and has been well characterized in PBL parameterizations (e.g., (Han and Bretherton, 2019)). In the context of convective boundary layer parameterization, the eddy-diffusivity (ED) approach based on K-theory (as discussed in Sect. 3.1) has been successful in representing neutral and surface boundary layers that promote more homogeneous and isotropic turbulence (Fig. 4-a). On the other hand, the flow heterogeneity induced by convection from thermal forcing and moisture phase change has been successfully captured by the mass-flux (MF) approach (Siebesma et al., 2007). In the effort to model both atmospheric conditions coexisting in PBL, an Eddy-diffusivity mass-flux (EDMF) scheme has been developed by partitioning flow fluctuations into contributions from local turbulent mixing (ED) and non-local coherent rising and descending air parcels (MF) (Lopez-Gomez et al., 2020; Lu, 2019).

This section explores this alternative approach to characterize the dispersive momentum flux other than a pragmatic approach developed in (Nazarian et al., 2020; Simón-moral et al., 2016) by considering dispersive transports as mass-flux components under the framework of the EDMF scheme. The dispersive transports of momentum (Fig. 4-b) in the urban boundary layer are induced by coherent motions near the building facets and present a strong local distribution that resembles the patchy coherent structures induced by thermal forcing in PBL. Therefore, the similarity in flow heterogeneity between UCL and PBL implies a phenomenological analogy, which favors the introduction of the EDMF approach to multi-layer UCMs. Following the formulation in (Siebesma et al., 2007), the dispersive flux of momentum is firstly decomposed into individual plumes ($a_i$) enclosing coherent motions with strength $(\overline{u_i} - \langle\overline{u}\rangle)(\overline{w_i} - \langle\overline{w}\rangle)$ and then sorted into upward (positive, with a fraction of $a_u$) and downward (negative, with a fraction of $a_d$) contributions.

$$\langle\tilde{u}\tilde{w}\rangle = \sum_i a_i (\overline{u_i} - \langle\overline{u}\rangle)(\overline{w_i} - \langle\overline{w}\rangle) = a_u\langle\tilde{u}\tilde{w}\rangle_u + a_d\langle\tilde{u}\tilde{w}\rangle_d \quad a_u + a_d = 1 \tag{8}$$

Note that, for simplicity, the entrainment process (Abdella and Petersen, 2000) that represents scalar transport between subdomains (between dispersive structures and to the turbulent environment) is not considered here. Using the MF approach to characterize dispersive momentum flux requires the fraction ($a_u + a_d = 1$) and strength ($\langle\tilde{u}\tilde{w}\rangle_u$, $\langle\tilde{u}\tilde{w}\rangle_d$) of those coherent flow motions to be parameterized based on the variation of urban arrangement. An example of the spatial distribution of turbulent and dispersive momentum flux is shown in Fig. 4 for the S-highrise-SD56 configuration of density $\lambda_p = 0.25$ sampled at $z = 15$m.





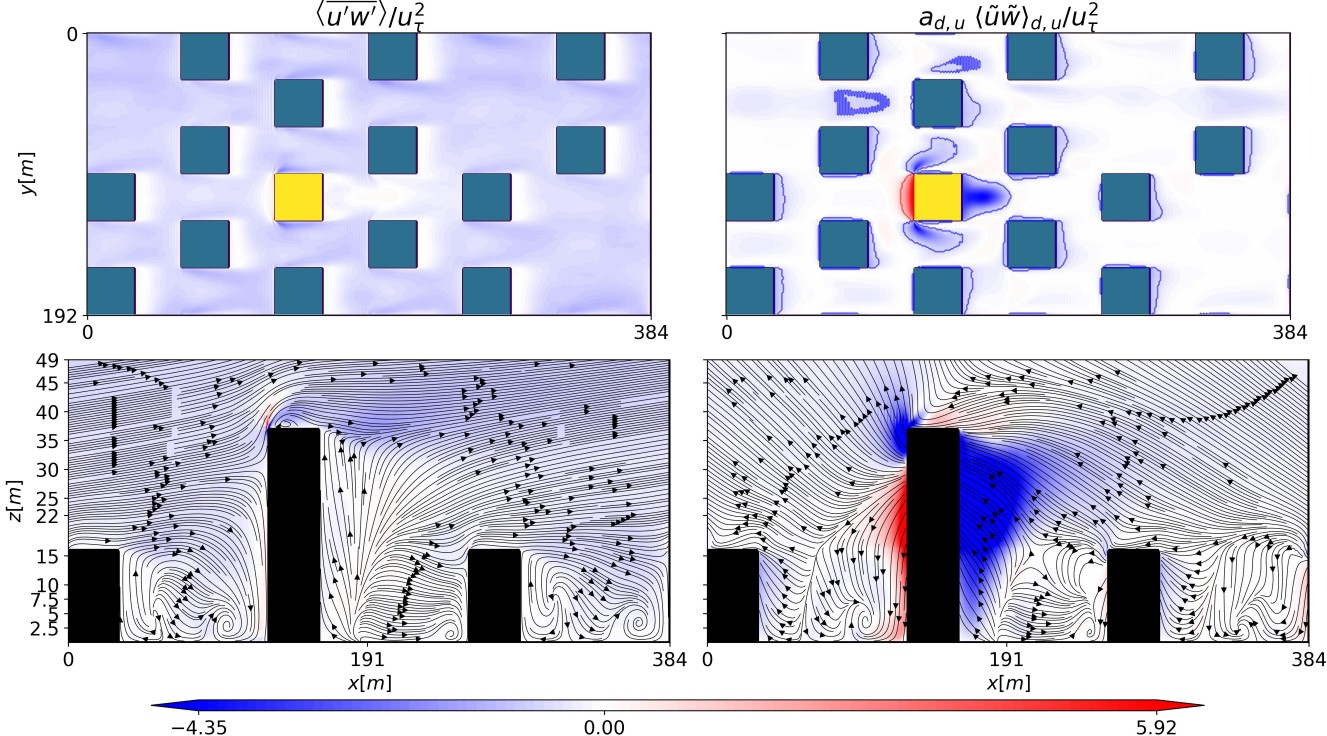

**Figure 4.** An example of horizontal and vertical field sampled at 93.8% of the mean building height ($z$ =15m) and $y = 110$m, respectively, for the S-highrise-SD56 configuration of density $\lambda_p = 0.25$. The left panel (a, c) shows the turbulent flux of momentum (a, $\langle \overline{u'w'} \rangle /u_\tau^2$=-0.6052) that forms the "Background" transport events, with the title showing the spatial average. The right panel (b, d) shows the dispersive flux that forms "hot-spot" transport events (b, $a_d \langle \tilde{u}\tilde{w} \rangle_d /u_\tau^2$=-0.2122, $a_u \langle \tilde{u}\tilde{w} \rangle_u /u_\tau^2$=0.0253), with the contribution from upward and downward transport indicated in the title. Regions enclosed by red and blue lines indicate values sampled with a 50% CDF. Vectors in the bottom left (c) panel show the mean velocity direction, whereas in the bottom right, they show the direction of the dispersive velocity.

Unlike the turbulent momentum flux, which shows a uniform downward transport with less correlation with the local rough-
ness elements, only a portion of the dispersive counterpart is directly connected to the local roughness, whereas the rest remains undisturbed. Therefore, a proper sampling filter should be applied before gathering flow properties for parameterization. Again, we adopt a common sampling approach in the development of the EDMF (Couvreux et al., 2010; Sušelj et al., 2012) based on the joint cumulative probability density function (CDF) of the flux ($\tilde{u}, \tilde{w}$).

$$\text{CDF}((\tilde{u}, \tilde{w})) = \int\limits_{\tilde{u}_{\min}}^{\tilde{u}_{\max}} \int\limits_{\tilde{w}_{\min}}^{\tilde{w}_{\max}} \text{PDF}(\tilde{u}, \tilde{w}) d\tilde{u} d\tilde{w} = 1 \tag{9}$$

The CDF can be adjusted from $[0, 1]$ that corresponds to different integration ranges of dispersive streamwise and vertical velocity (Eq. 9). By lowering the CDF, the sampled regions are adjusted smaller only to enclose regions with outstanding



strength. We tested a range of cut-off values CDF $\in [0.3, 0.4, 0.5, 0.6, 0.7]$ (test results are shown in the appendix and supplementary file), and the optimum cut-off value here is determined as $0.5$ for all cases considered. Figure 4-b shows the sampled upward and downward transports enclosed by red and blue lines, respectively.

The sampling filter aims to capture the non-Gaussian component of the dispersive momentum flux that the K-theory cannot characterize. Here, we test the correlation between dispersive streamwise velocity ($\tilde{u}$) and vertical velocity ($\tilde{w}$) to show how the urban structures modify the statistical distribution of dispersive fluxes and the corresponding characterization fulfilled by the sampling. Figure 5 shows the 2D CDF distribution for S-highrise-SD56 configuration of density $\lambda_p = 0.25$. The distribution exhibits a more eccentric behavior along the vertical range, with,

1. One tail growing stronger and thinner at the third quadrant ($\tilde{u} < 0, \tilde{w} < 0$) representing the blockage effects of the frontal area of the roughness (Fig. 4-d), which is well-captured by the sampling strategy (red squares in Fig. 5). The height distribution of building blocks can explain the thinner trend and reflect how the dispersive flux responds to the urban vertical structure.

   2. A relatively symmetric elongated region at the second ($\tilde{u} < 0, \tilde{w} > 0$) and fourth quadrant ($\tilde{u} > 0, \tilde{w} < 0$) representing
the leeward flow cavity (Fig. 4-b, characterized by the sampled flux at most heights) and flow divergence at the lateral facets (represented by the sampled flux at 2.5m), respectively. The elongation of both sides reflects a more irregular distribution of downward dispersive flux with heights, which is consistent with the analysis in (Lu et al., 2023a).

The previous approach (Nazarian et al., 2020) lumping the dispersive flux with its turbulent counterpart implicitly assumes it retains a Gaussian distribution, which is not representative in the present study. Only after subtracting values from pronounced
regions from the rest of relatively uniform regions with Gaussian distribution one may safely resort back to the lumping approach and update Eq. 8 with Eq. 10.

$$\langle \tilde{u}\tilde{w} \rangle = (a_u \langle \tilde{u}\tilde{w} \rangle_u)_{co=0.5} + (a_d \langle \tilde{u}\tilde{w} \rangle_d)_{co=0.5} + (\langle \tilde{u}\tilde{w} \rangle)_{\mathrm{residue}} \qquad (10)$$

We speculate the reason that dispersive flux has been long ill-represented in UCMs is attributed to the inappropriate scaling with the turbulent flux (e.g., (Poggi and Katul, 2008b)), which usually yields a small fraction without recognizing these two
processes are controlled by very different physical processes. Alternatively, the dispersive perturbations are caused by the pressure fields associated with the topography or the changed resistance to flow through the canopy (Finnigan et al., 2015; Finnigan and Shaw, 2008), which implies a possibility of scaling its strength to the external pressure gradient. This is only made possible recently due to the increased computational power and awareness of intrinsic urban heterogeneity (Boeing, 2019; Lu et al., 2023b). Therefore, we hypothesize the strength of sampled flow motions can be parameterized by pressure
gradient and urban morphological parameters as Eq. 11.

$$a_{u,d} \langle \tilde{u}\tilde{w} \rangle_{u,d} = f\left( z, \frac{\partial \overline{p}}{\partial x}, \lambda_p, H_{\mathrm{max,mean,std}} \right), \qquad (11)$$



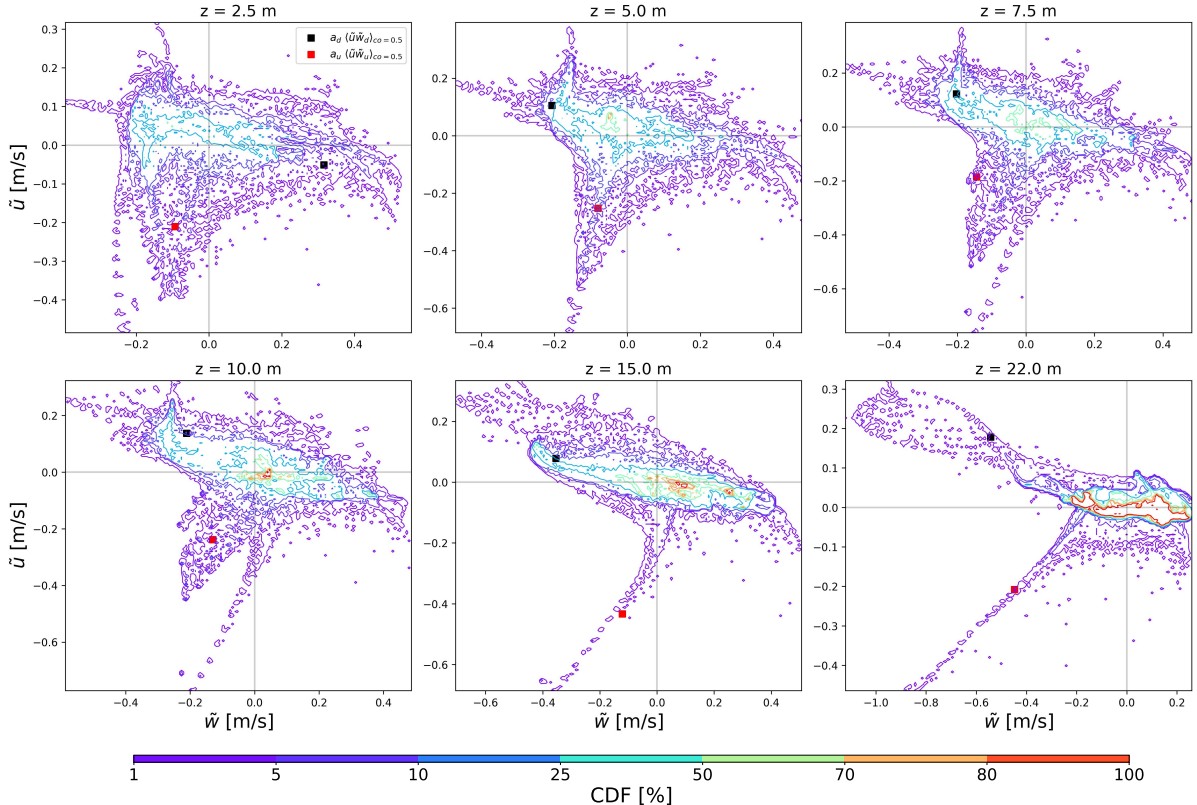

**Figure 5.** 2D CDF of the dispersive velocities at 2.5, 5, 7.5, 10, 15, and 22 m. The integral of the PDF within the area is delineated by isolines on the logarithmic scale. The black/red squares depict the averaged dispersive flux over the sampled regions to represent the strength of the gradient and counter-gradient transport that will be parameterized in MLUCM v3.0.

To gain a direct impression of how the sampled dispersive motions scale with the pressure gradient, we present the profile of the vertical gradient of the sum of two sampled terms from Eq. 10 in Fig. 6. The variation of the dispersive momentum transport exhibits a strong dependence on the flow regime but decouples with the turbulence level (not shown here; see Fig. 9-11 from (Lu et al., 2023b) for spatial pattern of TKE, turbulent and dispersive momentum flux). The sampled dispersive motion is of a similar magnitude to the only driver in our experiments - pressure gradient collectively leaving such a term unattended is equivalent to only applying half of the pressure gradient to the flow and inevitably degrades model predictability. Considering that realistic urban geometries will exhibit increased complexity and dispersive flux (Giometto et al., 2016), explicit parameterization is necessary. The little variation of profiles for each density in Fig. 6 indicates building height distribution does not induce significant variance over the sampled dispersive flux among cases within the $H_{\mathrm{mean}}$. Instead, urban density dictates the variation of the sampled flux, whose profile is different between sparse (the first row in Fig. 6) and denser (the second row in Fig. 6) layouts: For the three sparse layouts, a relatively linear increasing trend crossing zero with heights is developed. Denser layouts lead to strong overlapping of building wakes, which helps to maintain a constant strength profile of similar



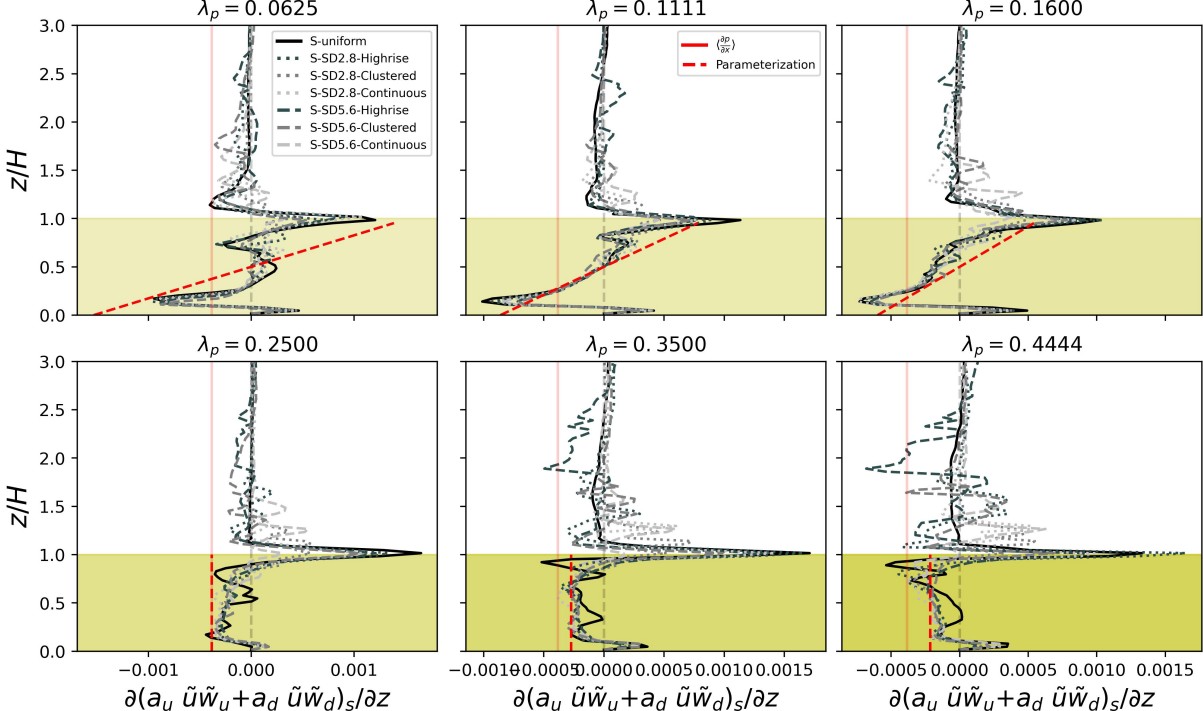

**Figure 6.** Vertical profiles of sampled downward and upward dispersive momentum flux (grey-scale lines) for seven building height configurations covering $\lambda_p \in [0.0625, 0.4444]$. The range of mean building height is marked in yellow colors with the transparency following the $\lambda_p$. Diminished red lines indicate the strength of the external pressure gradient constant with height. Red dash lines show the parameterization from Eq. 12.

magnitude to the pressure gradient, which indicates a trend of skimming flow regime (Oke et al., 2017). Above the $H_{\mathrm{mean}}$,

sampled fluxes quickly converge to zero except for the highrise configurations over denser layouts ($\lambda_p \leq 0.25$) and at the $H_{\mathrm{max}}$ of each configuration, demonstrating a minor magnitude but complex pattern across configurations. Therefore, for simplicity, we only provide a parameterization for the sampled fluxes below $H_{\mathrm{mean}}$ that scales with the external pressure gradient $\frac{dp}{dx}$ as follows,

$$\frac{d\left((a_d \langle \tilde{u}\tilde{w} \rangle_d)_{co=0.5} + (a_u \langle \tilde{u}\tilde{w} \rangle_u)_{co=0.5}\right)}{dz} = \frac{H_{\mathrm{mean}}-2z}{H_{\mathrm{mean}}} \frac{0.25}{\lambda_p} \left| \frac{\partial \overline{p}}{\partial x} \right| \quad \lambda_p < 0.25 \tag{12a}$$

$$= \frac{0.25}{\lambda_p} \left| \frac{\partial \overline{p}}{\partial x} \right| \quad \lambda_p \geq 0.25 \tag{12b}$$

The above parameterization retains the physical significance of upward and downward dispersive motion demonstrated in Fig. 6. By definition, the scaling coefficients to the external pressure gradient are designed to be lowered over denser layouts, consistent with previous findings where dispersive momentum flux is more significant over sparse arrangements (e.g., (Lu et al., 2023b)).





### 3.3 Drag and dissipation parameterization

The presence of roughness elements mounted on the urban surface implicitly modifies turbulent length scales (Li et al., 2020) that regulate transport mechanisms of flow properties. Explicitly, the buildings block and divert the flow, inducing pressure deficits between their windward and leeward facets that collectively serve as a sink for momentum (Eq. 1) and source for TKE (Eq. 4). In the following, we will revisit the modeling strategy for drag and dissipation in Nazarian et al. (2020) and add necessary adjustments to reflect the building height variability. By assuming the strength of building effects is proportional to the area facing the wind per cubic meter of outdoor air volume $S(z)$ and the spatially averaged mean wind speed $\langle \overline{U} \rangle$ and a drag coefficient $C_d$ encodes the variation of building arrangements, the drag parameterization reads,

$$\frac{1}{\rho}\left\langle \frac{\partial \bar{P}}{\partial x} \right\rangle \bigg|_z = S(z)C_d\langle \overline{u(z)} \rangle |\langle \overline{u(z)} \rangle|, \tag{13a}$$

$$-\left\langle \widetilde{u_i'u_j'}\frac{\partial \tilde{u}_i}{\partial x_j} \right\rangle + D_k = S(z)C_d|\langle \overline{u(z)} \rangle|^3. \tag{13b}$$

Accordingly, characterizing drag impacts converts to evaluating the drag coefficient $C_d$ that comes in different forms appearing in the literature (e.g., (Santiago et al., 2013)). The equivalent drag coefficient $C_{\text{deq}}$ evaluated following Santiago and Martilli (2010b) requires fewer inputs but maintains similar performance demonstrated its capacity in MLUCM v2.0 (Nazarian et al., 2020). It will be retained in the present study. Using $C_{\text{deq}}$ to evaluate assumes a constant profile of drag that ensures the integral of the drag over the whole urban canopy is equivalent to that evaluated from LES simulations.

$$C_{\text{deq}} = \frac{\frac{-1}{\rho H}\int_0^H \Delta\langle \overline{p(z)} \rangle \mathrm{d}z}{\frac{1}{H}\int_0^H \langle \overline{u(z)} \rangle |\langle \overline{u(z)} \rangle| \mathrm{d}z} \tag{14}$$

Where $\Delta\langle \overline{p(z)} \rangle = p_f(z) - p_b(z)$ is the pressure deficit between the windward and leeward facets of each building block. Due to the low variance $C_{\text{deq}}$ over the impact of height variability except for denser layouts $\lambda_p \geq 0.35$, over each density in table 2, we binned all 7 cases to show their mean and standard deviation in Fig. 7 together with that from (Nazarian et al., 2020) for uniform staggered arrays. The excellent agreement with $C_{\text{deq}}$ in the present study indicates the parameterization in (Nazarian et al., 2020) presents great resilience to the height variability, especially over the sparse layouts. Therefore, updating the constants in the original parameterization requires only a minor change. However, height variability not only extends the vertical range of the urban canopy but also complicates the vertical structure of the drag coefficient and requires extra characterization. Figure 7-b shows the ratio between the drag coefficient above the mean building height $H_{\text{mean}}$ and $C_{\text{deq}}$ that covers a range that is generally larger than unity and shows a decreasing trend with densities that can be captured by considering the morphological condition of the urban surface from (Lu et al., 2023a).

The last term parameterized is the dissipation length scale from Eq. 5. As indicated from the left panel of Fig. 8, the impact of height variability $l_\varepsilon/C_\varepsilon$ below the mean building height only demonstrates a very minor variation for each density range. The variation is more evident above $H_{\text{mean}}$ with a clear shape following the sectional density and was discussed in (Lu



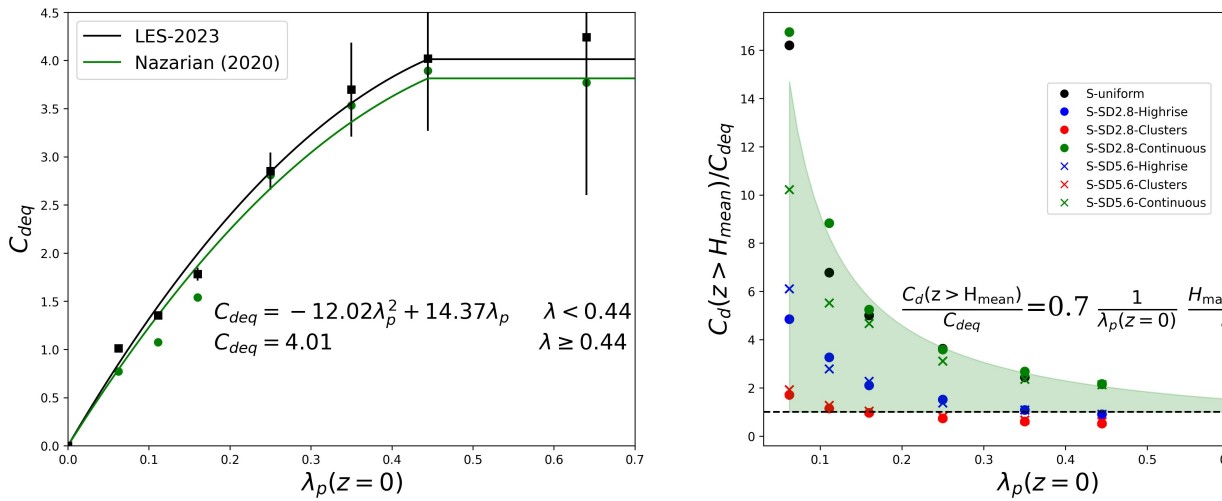

**Figure 7.** Drag parameterization based on the equivalent drag coefficient $C_{\mathrm{deq}}$. The left panel shows the errorbar plot where black squares mark the binned value of $C_{\mathrm{deq}}$ among seven building height configurations, and the vertical line attached to each indicates the standard deviation for each bin. The green dots indicate $C_{\mathrm{deq}}$ evaluated over uniform-staggered cases from (Nazarian et al., 2020). The right panel shows the scatter plot of the ratio between $C_{\mathrm{deq}}$ and the drag coefficient evaluated above the mean building height where the dashed line indicates the ratio of 1. The green shaded area indicates the corresponding parameterization in the plot.

et al., 2023a). Despite applying a sophisticated dissipation and drag parametrization might lead to better model performance
(Glazunov et al., 2021; Castro, 2017; Sützl et al., 2021a), it is beyond the scope of the present study focused on refining the transport characterization of UCM. Therefore, we decided to keep the original parameterization from MLUCM v2.0 (Nazarian et al., 2020). For completeness, we briefly describe the parameterization strategy: By segmenting the urban canopy flow into the mixing layer, transition region, and wall-bounded layer (Coceal and Belcher, 2004), the parameterization reads,

$$l_\varepsilon/C_\varepsilon = \alpha_1(H_{\mathrm{mean}} - d) \quad z/H_{\mathrm{mean}} < 1, \tag{15a}$$

$$l_\varepsilon/C_\varepsilon = \alpha_1(z - d) \quad 1 \leq z/H_{\mathrm{mean}} \leq 1.5, \tag{15b}$$

$$l_\varepsilon/C_\varepsilon = \alpha_2(z - d_2) \quad z/H_{\mathrm{mean}} > 1.5, \tag{15c}$$

Where $\alpha_1 = 5.5$ is a revised value based on 49 simulations in the present study. $d = H_{\mathrm{mean}}\lambda_p^{0.15}$ is the displacement height parameterized following Krayenhoff et al. (2015). Eq. 15-a depicts a mixing layer type of flow with a constant dissipation efficiency, Eq. 15-b represents the transition to a fully wall-bounded flow Eq. 15-c where the dissipation efficiency decreases
with height based on $d_2$ and $\alpha_2$ parameterized as follows,





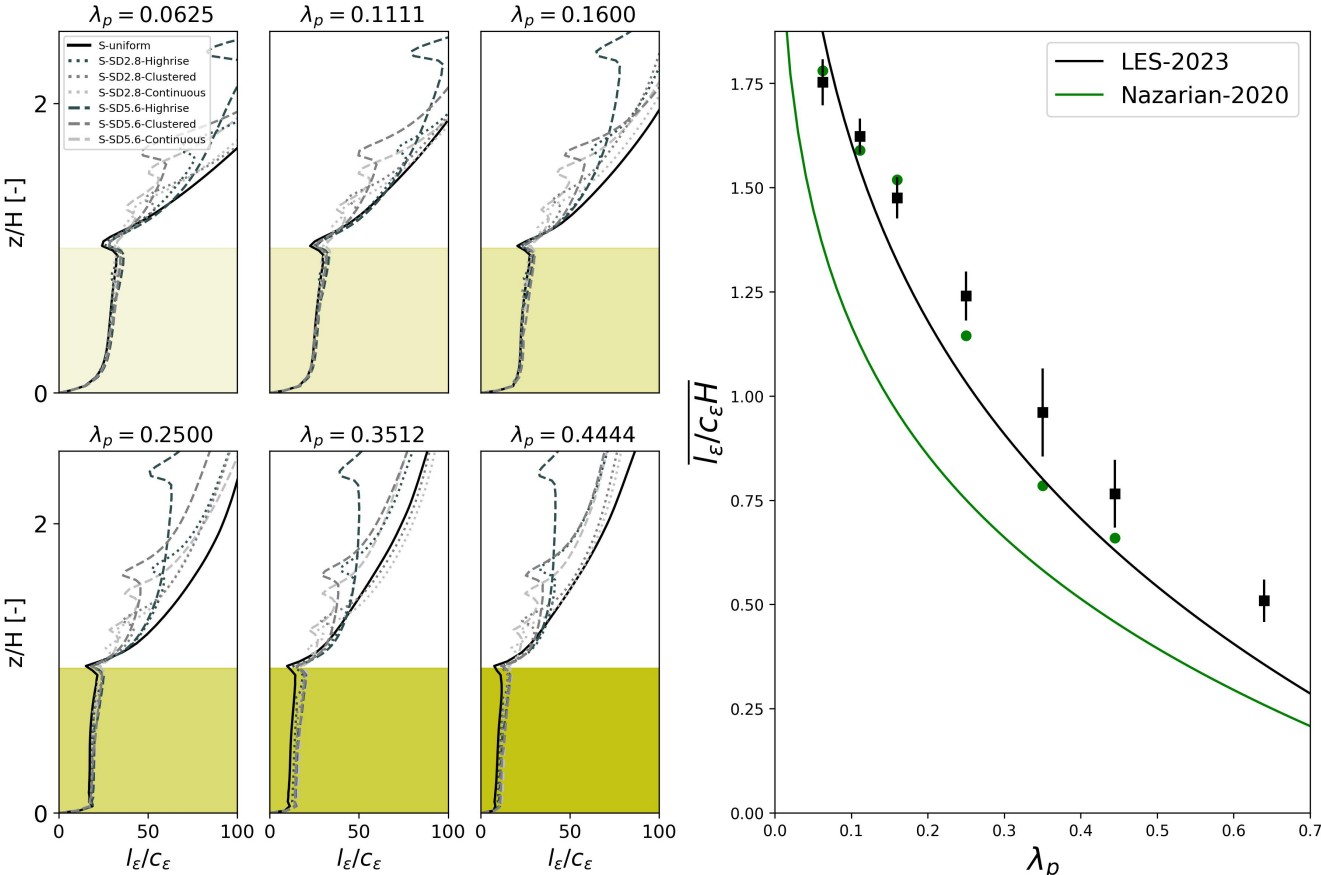

**Figure 8.** The left panel shows the vertical profiles of dissipation length scales $l_\varepsilon/C_\varepsilon$ up to $2.5H_{\mathrm{mean}}$ where the yellow-shaded range indicates mean building height. The right panel shows the binned vertical averaged dissipation length scale $\overline{l_\varepsilon/C_\varepsilon}$ following the same manner as Fig. 7 overlaid with the parameterization and with that from Nazarian et al. (2020).

$$\alpha_2\left(\lambda_{\mathrm{p}}\right) = \min\left(5, \max\left(2, 1.3\lambda_{\mathrm{p}}^{-0.45}\right)\right) \tag{16a}$$

$$d_2\left(\lambda_{\mathrm{p}}\right) = 1.5H\left(1 - \frac{\alpha_1}{\alpha_2}\right) + d\left(\lambda_{\mathrm{p}}\right)\frac{\alpha_1}{\alpha_2} \tag{16b}$$

## 4  MLUCM performance with updated parameterizations

In this section, we evaluate the predictability of two modifications to the original MLUCM developed in Nazarian et al. (2020), where the combination of changes proposed in Sect. 3.1 and Sect. 3.2 and the original model forms four scenarios, as shown in Table 2.





**Table 2.** Four scenarios tested in the present study with the combination of EDMF approach (Sect. 3.2) and differentiation on the TKE diffusion (Sect. 3.1 ). In the table, **1D-Original** aims to set a baseline by employing a similar parameterization from Nazarian et al. (2020) with an adaptation for the height variability from Sect. 3.3. **1D-$K_k$** only considers the differentiation of TKE transport from momentum as discussed in Sect. 3.1. In contrast, **1D-EDMF** only considers the "EDMF" parameterization for dispersive momentum flux while **1D-$K_k$EDMF** considers both modifications.

| Scenario | $K_m$ | $K_k$ | Dispersive momentum flux |
|---|---|---|---|
| **1D-Original** | $\langle \overline{u'w'} \rangle + \langle \tilde{u}\tilde{w} \rangle$ | $\langle \overline{u'w'} \rangle + \langle \tilde{u}\tilde{w} \rangle$ | To $K_m$ |
| **1D-$K_k$** | $\langle \overline{u'w'} \rangle + \langle \tilde{u}\tilde{w} \rangle$ | $\langle \overline{\boldsymbol{k'w'}} \rangle + \langle \tilde{\boldsymbol{k}}\tilde{\boldsymbol{w}} \rangle$ | To $K_m$ |
| **1D-EDMF** | $\langle \overline{\boldsymbol{u'w'}} \rangle + \langle \tilde{\boldsymbol{u}}\tilde{\boldsymbol{w}} \rangle_{\mathbf{residue}}$ | $\langle \overline{u'w'} \rangle + \langle \tilde{u}\tilde{w} \rangle$ | EDMF |
| **1D-$K_k$EDMF** | $\langle \overline{\boldsymbol{u'w'}} \rangle + \langle \tilde{\boldsymbol{u}}\tilde{\boldsymbol{w}} \rangle_{\mathbf{residue}}$ | $\langle \overline{\boldsymbol{k'w'}} \rangle + \langle \tilde{\boldsymbol{k}}\tilde{\boldsymbol{w}} \rangle$ | EDMF |

Figure 9 shows the vertical profiles of horizontally averaged velocity, TKE, and turbulent momentum flux calculated with the 1D models with different transport characterization and compared with the corresponding LES results for the **High-rise** configuration with $H_{\mathrm{std}}$ =5.6m. Profile comparison for other configurations can be found in the supplementary file. We se-

lect sparse ($\lambda_p$=0.1111), medium ($\lambda_p$=0.25), and dense ($\lambda_p$=0.4444) layouts to reflect different flow regimes for comparison. Overall, scenarios do not present a great variation except for the in-canopy TKE profile. **1D-$K_k$** and **1D-$K_k$EDMF** with differentiation of $K_k$ from $K_m$ (Table 2) demonstrate a significant improvement on the TKE profile for medium and dense layouts below $H_{\mathrm{mean}}$.

The introduction of EDMF is phenomenologically equivalent to imposing a stronger pressure gradient, which leads to higher

velocity and explains the better agreement with LES for **1D-EDMF** and **1D-$K_k$EDMF** scenarios. As a result, **1D-$K_k$EDMF** yields an even higher TKE due to the introduction of an MF term for dispersive transport of momentum in Eq. 1 as an explicit source of momentum leading to higher TKE production. Correction on TKE not only leads to a better agreement on the velocity with LES but also helps mediate the overestimation of daytime air temperature by correcting the vertical exchange of heat (Krayenhoff, 2014).

The variation of performance of scenarios is not pronounced across different height variability, which further consolidates the binning strategy for parameterization in the present study. To evaluate the comprehensive performance of scenarios over different height configurations, Fig. 10 presents the root mean square error (RMSE) between vertical profiles (ranging from $z = [0 - 3H_{\mathrm{mean}}]$) of LES and four MLUCM scenarios averaged over seven configurations (one uniform and six variable, as shown in Table 1) over seven densities.

The new parameterizations with refined transport characterization represent an overall improvement compared to the previous multi-layer model (Nazarian et al., 2020). All three scenarios with modifications lead to better predictability for velocity except for the two sparse layouts. The predictability for TKE is substantially better from **1D-$K_k$** and **1D-$K_k$EDMF** scenarios except for two sparse layouts that can be further improved with a better characterization on the TKE budget terms such as dissipation and wake production (Blunn et al., 2022). The performance on turbulent momentum flux ($\langle \overline{u'w'} \rangle / u_\tau^2$) is generally







**Figure 9.** Vertical profiles of velocity ($\langle \overline{u} \rangle / u_\tau$), turbulent kinetic energy ($\langle \overline{k} \rangle / u_\tau^2$), and turbulent momentum flux ($\langle \overline{u'w'} \rangle / u_\tau^2$) obtained with the four scenarios in Table 2 and LES results for sparse ($\lambda_p =0.1111$), medium ($\lambda_p =0.25$), and dense ($\lambda_p =0.4444$) layouts.

similar across scenarios except for **1D-EDMF** and **1D-$K_k$EDMF** being superior in sparse layouts. Having the most remarkable performance, **1D-$K_k$EDMF** with both modifications from Sect. 3.1 and Sect. 3.2 is desirable for MLUCM v3.0.



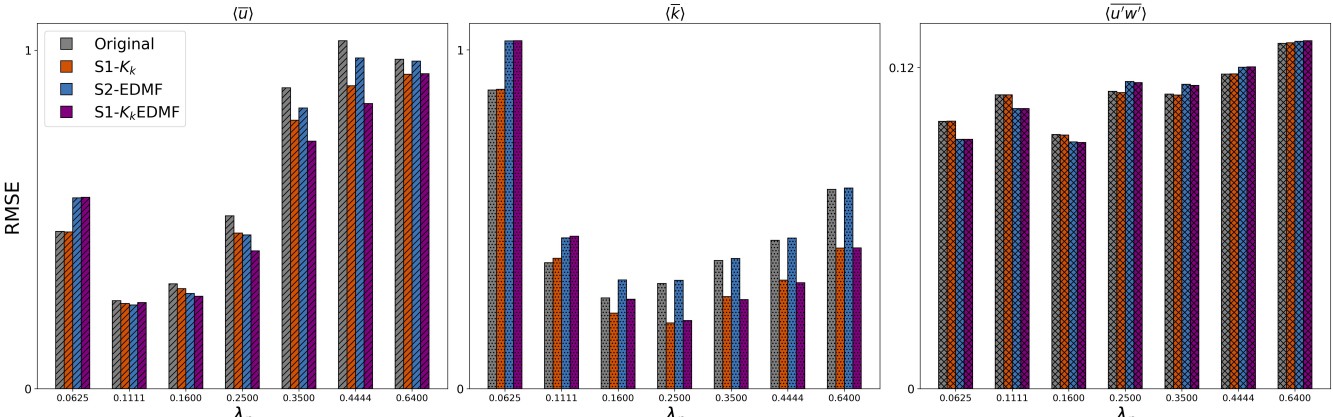

**Figure 10.** RMSE for four different scenarios of the 1D model from Table 2 against LES across seven densities in combination with seven height configurations from Table 1. Each column shows the lumped RMSE across seven different configurations.

## 5    Summary and conclusions

This refined the characterization of the transport of flow properties for the multi-layer urban canopy model via a separate parameterization of TKE diffusion and introduced a "mass-flux" term for the dispersive transport of momentum. The updated multi-layer model demonstrates improved performance by correcting the underestimation of turbulent exchange and velocity below the mean building height $H_{\mathrm{mean}}$.

Through analyzing 49 LES simulations over staggered urban arrays of uniform and variable building height and comprehensive density coverage, we found the turbulent exchange rate of momentum is similar to that for scalars but 3.5 times lower for TKE. However, the previous model (MLUCM v.2.0, (Nazarian et al., 2020)) assumed a unified transport efficiency for momentum and TKE, which curbs the transport of the high TKE flow above $H_{\mathrm{mean}}$ into the canopy and causes underestimation of in-canopy TKE and wind speed. The accurate characterization of TKE transport into the lower canopy becomes critical as it also controls the turbulent exchange of heat and moisture from buildings and vegetation such as BEP-Tree. For example, underprediction of canopy TKE leads to overprediction of daytime air temperature and diurnal temperature range (Krayenhoff et al., 2020). Therefore, the updated eddy diffusivity for TKE incorporated in MLUCM v3.0 is expected to benefit BEP-Tree and multi-layer urban meteorology models broadly.

We also revealed the non-Gaussian nature of the horizontal distribution of dispersive momentum flux, which is induced by flow heterogeneity responding to windward (quadrant 3, upward), lateral (quadrant 4, downward), and leeward (quadrant 2, downward) flow patterns. In response to this complication, we applied a sampling filter to the cumulative density function of the dispersive transport to segment the non-Gaussian contribution (MF) that scales well with the pressure gradient and the Gaussian component that can be securely lumped into the turbulent counterpart in the 1.5-order turbulent closure (ED). The EDMF framework, having successfully modeled the planetary boundary layer with flow heterogeneity induced by thermal



effects, was demonstrated to be also favorable for the representation of flow heterogeneity caused by rigid building volumes. The new framework enables an explicit consideration of the impact of flow heterogeneity in MLUCM, improving our ability to calibrate the model for simulations of complex canopy flow.

With adaptations to the parameterizations of drag coefficient and dissipation length scale to account for building height variability, the model configuration that includes both updated eddy-diffusivity of TKE ($K_k$) and the EDMF scheme yields an improved performance relative to MLUCM v2.0. However, analysis of the flow profiles and length scales indicates further investigation is needed to address the following aspects:

1. The turbulent length scales show a great variance over heights above $H_{\mathrm{mean}}$ for layouts with non-uniform building height distribution. A height-dependent parameterization of turbulent length scale and drag profile is necessary to correctly simulate the logarithmic and exponential behavior of wind speed profile (Castro, 2017).

2. Following the dispersive momentum flux parameterization, the dispersive transport of TKE and scalar can be considered in a similar manner where the sampling criteria may not be universal. In the original implementation of EDMF, the sampled air parcel also exchanges flow properties to the turbulent environment and other air parcels, which could be considered in future developments.

3. Staggered arrays were found to be less representative for flow over realistic dense urban neighborhoods (Lu et al., 2023b) that also respond to wind directions (Santiago et al., 2013) and thermal effects (Simón-moral et al., 2016). Further simulation for model calibration should account for these factors for a more realistic flow characterization.

4. Similar to the scarcity of evaluation on the eddy-diffusivity of flow properties other than momentum, the variation of TKE budget terms is still poorly considered in urban flow models. Accordingly, a more comprehensive analysis that addresses relationships between the urban morphology and the TKE budget may provide further improvements.

*Acknowledgements.* This research was supported by the Australian Research Council Centre of Excellence for Climate Extremes (Grant CE170100023). Simulations were undertaken with the assistance of resources and services from the Australian Government's National Collaborative Research Infrastructure Strategy (NCRIS), with access to computational resources provided by the National Computational Infrastructure (NCI), which is supported by the Australian Government through the National Computational Merit Allocation Scheme. We also thank the anonymous reviewers for their constructive and valuable comments.




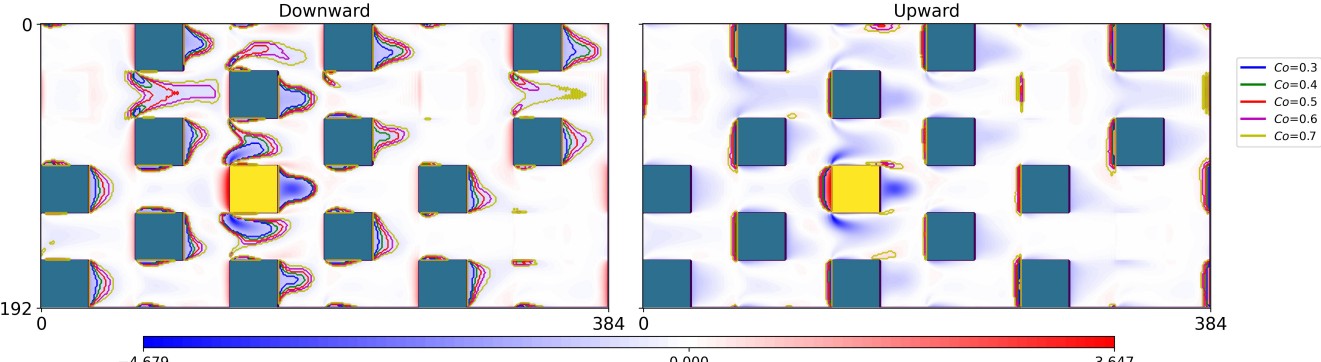

**Figure A1.** Sampling regions of dispersive fluxes for five ($co \in [0.3, 0.4, 0.5, 0.6, 0.7]$) cut-off CDF values.

## Appendix: Sampling of the dispersive momentum flux

The sampling procedure to extract the non-Gaussian contribution of dispersive fluxes is designed as follows: a) Separating the dispersive fluxes into gradient (downward) and counter-gradient (upward) components; b) Sorting these fluxes by their gradients (from large to small) and lowering the cumulative density function of the two fluxes from the largest magnitude from 1 allows a smaller subdomain to be sampled until subdomains only enclose transport events connected to buildings; c) Spatially averaging the dispersive fluxes over subdomains for both gradient and counter-gradient DMF. Figure A1 shows a range of cut-off values ($co \in [0.3, 0.4, 0.5, 0.6, 0.7]$) for CDF over an example for $\lambda_p$=0.25. The optimum cut-off value here is determined as 0.5 for all cases considered (an example is provided in Fig. 4-b, where the sampled downward(upward) transports are enclosed by blue(red) lines).

The 3D distribution of the sampled dispersive structures is shown in Fig. A2 S-Highrise-SD56 for $\lambda_p \in [0.0625, 0.4444]$ where the dispersive momentum flux concentrates in the immediate vicinity downstream of the cubes. Structures near the ground contributed significantly to negative stresses (green structures) where the recirculating flows dominate. In response to this distribution (note the dispersive velocity vectors in Fig. 4-d)), the parameterization for the vertical variation of sampled dispersive fluxes is stronger over the lower part of the canopy over sparse layouts, which is in general agreement with Poggi and Katul (2008a). In the upper levels of the canopy, the counter-gradient stresses (red structures) are more pronounced at the windward facets due to flow reflection.

*Code availability.* The source code and the supporting data of three scenarios for the 1D Multi-layer Urban Canopy Model are publicly available at https://doi.org/10.5281/zenodo.10207052 (last access: Nov 2023) under GPL 3.0 license.

*Data availability.* Dataset is available in the supplementary files



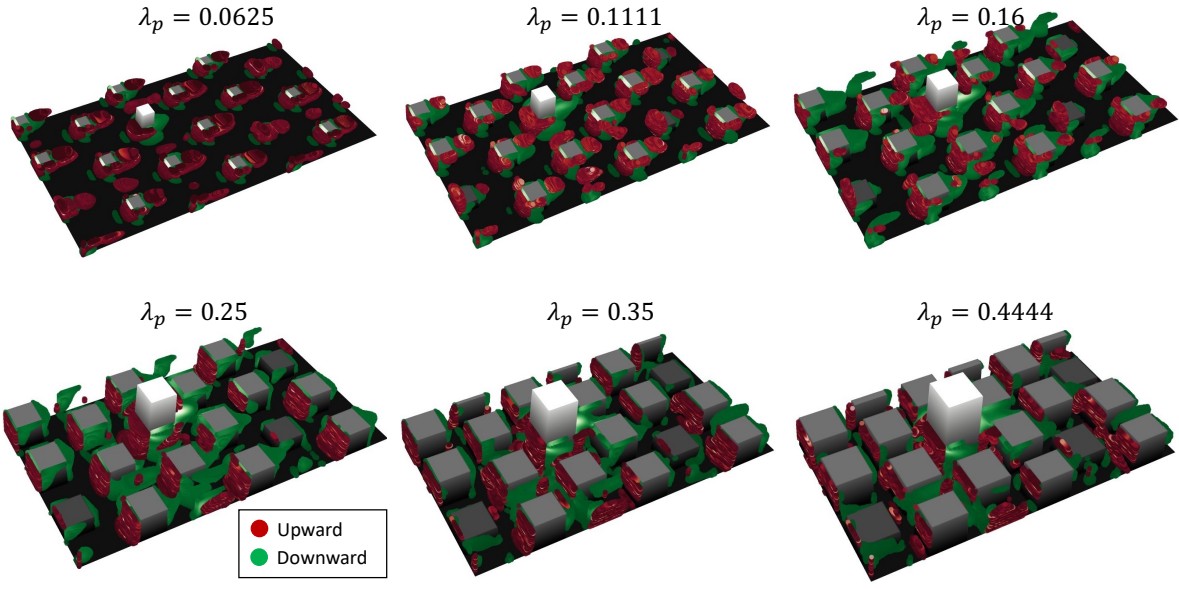

**Figure A2.** 3D distribution of the dispersive coherent structures for cases with S-Highrise-SD56 covering $\lambda_p \in [0.0625, 0.4444]$. Green and red regions represent gradients and counter-gradient dispersive transport sampled with a 50% CDF.

*Author contributions.* JL, NN, MH, ESK, and AM collectively developed and planned the study. JL ran the large-eddy simulations designed by NN and modified the model with the help of NN and AM. JL carried out the result analyses and wrote the paper with significant input and critical feedback from NN, AH, ESK, and AM.

*Competing interests.* The authors declare that they have no conflict of interest.

*Acknowledgements.* This research was supported by the Australian Research Council Centre of Excellence for Climate Extremes (Grant CE170100023). Simulations were undertaken with the assistance of resources and services from the Australian Government's National Collaborative Research Infrastructure Strategy (NCRIS), with access to computational resources provided by the National Computational Infrastructure (NCI), which is supported by the Australian Government through the National Computational Merit Allocation Scheme. We also thank the anonymous reviewers for their constructive and valuable comments.



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
