# Peer review of "A one-dimensional urban flow model with an Eddy-diffusivity Mass-flux (EDMF) scheme and refined turbulent transport (MLUCM v3.0)"

_EGUsphere, 2023_

## Author Comment (AC1)

Dear Editor,

Please find below the detailed response to the reviewers' comments and suggestions. We thank the reviewers for their detailed assessment of our work and believe that the quality of the manuscript has improved based on their feedback.

We hope that the revised manuscript meets the standards for publication in the Geoscientific Model Development and look forward to your feedback.

Best,
Jiachen Lu (On behalf of all co-authors)

Note for coloring and formatting for the response:

Black font marks the comment from the reviewer.
Blue font marks the response.
*Blue italic font marks the corresponding revision in the manuscript.*
***Bold blue italic font marks the newly added contents in the manuscript.***

**Response to Reviewer #1**

The MS has a range of interesting results on flow and transport in urban terrain. The LES quality is excellent and the LES is used to inform parametrization in a widely used UCM model, the MLUCM. The results thus have the potential to contribute to advancing the field.
The paper is overall well written (some parts could be improved) and can be published with moderate modifications.

**Response:** We thank the reviewer for the detailed feedback on the manuscript. Please see below for responses to comments.

Major Comments
1) What models or application besides MLUCM could benefit from the developments of the upgraded closure and how? As is, the paper is presented as simply an effort to improve MLUCM, which misses the chance to reach a broader audience.

We thank the reviewer for the suggestion. We have added related applications besides MLUCM as a separate paragraph in the conclusion section to attract interest from a broader audience.

***Regarding applications, the enhanced model, which addresses the underprediction of in-canopy TKE, has the potential to alleviate the overestimation of daytime air temperature and diurnal temperature range (Krayenhoff et al., 2020). Furthermore, the improved prediction of urban fluxes reinforces the advantages of MLUCM over simpler bulk schemes, emphasizing their ability to realistically simulate vertical diffusion processes in mesoscale atmospheric models Hendricks et al. (2020). The proposed refinements are not exclusively constrained to urban applications but are also applicable to broader turbulence modeling methods. For example, the explicit***

*parameterization of dispersive momentum flux could benefit turbulence modeling over other types of roughness elements such as coral reefs (Davis et al., 2021), and vegetative canopies (Finnigan, 2000). In parallel, emphasizing the distinction between TKE transport efficiency and its momentum counterpart underscores the importance of gathering higher flow moments for modeling general atmospheric flow (Ghonima et al., 2017).*

2 ) Line 111: LES solves the Navier Stokes equation with the Boussinesq approximation, not the Boussinesq equation. The Boussinesq equation is a different PDE that describe wave propagation.

Thanks to the reviewer for noticing this incorrect language. We have corrected the expression in the manuscript.

*The momentum field is solved using* **the filtered Navier Stokes equation with the Boussinesq approximation** *with the time integration following a minimal storage scheme*

3) I am a bit confused by the explanation of the terms in eqs. 1 and 2.

(i) The authors write "The fourth term of Eq. 1 represents a term risen from spatially averaging that accounts for momentum sink due to form and skin drag." This seems to relate to this term This is quite confusing since this looks like the mean pressure term. For it to be defined as drag, the P here should be defined as the perturbation from an otherwise linearly decreasing pressure in x. Why don't the authors just call this drag $D_i$ ?

Thanks to the reviewer for the suggestion. We prefer to term the building-induced drag explicitly as P here because the drag is directly evaluated from the **pressure** drop between windward and leeward building facets. For drag parameterization, we followed the "equivalent drag coefficient" from Eq. 26 from Santiago et al. (2010) and noted this choice From Eq. 13a in the manuscript. We have made this clear by revising the manuscript:

*The fourth term of Eq. 1 represents the pressure drop between windward and leeward building facets (Santiago and Martilli, 2010b), which accounts for momentum sink due to building-induced drag rising from spatially averaging over the momentum equation.*

(ii) the last term is the viscous stress which they never explain, and they omit the corresponding molecular flux term in eq. 2. LES at their Re numbers should not be including the viscous term so It is clearer to remove it.

Thanks to the reviewer for the comment, we have removed the viscous stress and added a sentence to explain the reason.

*The friction Reynolds number (Re $= \frac{u_\tau H_T}{\nu}$, where $u_\tau \sim 0.21m/s$ is the friction velocity, $H_T$ is domain height, and $\nu$ is the kinematic viscosity) of the flow dataset is large enough ( $10^6$)* **to neglect viscous stress on momentum and molecular flux on scalars.**

4) In various places the authors write "non-Gaussian dispersive momentum transport". Not sure why. It seems to distinguish them from a Gaussian turbulent transport, but the turbulent perturbations are not Gaussian either. Nothing here is Gaussian, so why this specification?

Thanks to the reviewer for the comment, indeed, Gaussian doesn't seem to be the most appropriate term here for the spatial variation of dispersive fluxes. Looking at Figure 5, we conceptualize the mass-flux parameterization here as an effort to extract the tails that represent dispersive structures adjacent to buildings, and the CDF/PDF of the rest should have a more circular-shaped distribution.

We have renamed "non-Gaussian" to "fraction of dispersive momentum flux directly induced by buildings" and "Gaussian" to "The relatively homogeneous component".

5) Equation 12: at steady state in an LES, the driving pressure gradient has to balance the surface drag (and Coriolis if present). So why not scale with the total surface drag instead?

We thank the reviewer for the valuable comment. In terms of coupling with mesoscale models, the pressure gradient represents one of the atmospheric forcings, which is more most relevant for this model than the total surface drag that focuses on the resistance of the urban surface.

6) related to 5: I suspect the minor influence of the model on the profiles of U and u'w' is because of the imposed global force balance. At any given height the stress divergence + pressure gradient (driving the flow) must balance building drag (slowing the flow). Since the building drag is imposed in 13a and the pressure gradient is also imposed, the stress divergence is also constrained. This is why the stress profiles in the rightmost column of Fig 9 are identical for all runs. This then also constrains -KdU/dz, and explain the small differences in the U profile.

TKE does not have such constraints and varies more. Maybe more importantly, the heat or scalar flux are also usually not constrained and could vary more with changing closures.

We thank the reviewer for the valuable comment noting the little variation of U and u'w' over different scenarios. However, the turbulent momentum flux is diagnostically obtained from MLUCM based on -KdU/dz, where K depends on TKE. In the present work, the force balance between pressure gradient and building drag is different from the conventional budget (e.g., in Sutzl et al. (2021)) because a mass-flux term is introduced as an explicit source of momentum. As a result, scenarios with EDMF have a higher velocity profile. We have added this explanation to make it clear:

*The prediction on streamwise velocity does not present a great variation where 1D-$K_k$-EDMF exhibits the highest value. This is explainable by the newly introduced MF term as an explicit source of momentum and differentiation of $K_k$ from $K_m$ enhancing the momentum transport from the atmosphere to the canopy.*

We envision future improvements on MLUCM should be in the turbulence closure itself (higher closure) or alternative scaling at regions where the model underperformed (Groud and canopy interface): We have revised the manuscript to make this clear in the conclusion section about potential improvements:

*The applicability of the 1.5-order k−l turbulence closure in MLUCM is restricted to isotropic turbulence flow, where the momentum exchange rate is directly correlated with turbulence strength. However, this scaling becomes questionable even in neutral conditions and diminishes under thermal stratification, where turbulence tends to be highly anisotropic (Stiperski and Calaf, 2018). In response to these limitations, employing a higher-order closure that diagnostically assesses momentum transfer (Wilson, 1988; Ayotte et al., 1999) or considering alternative scaling approaches for*

*poorly performing regions (Sun et al., 2020) may contribute to the development of more effective urban canopy models in the future.*

It's also our ongoing work to explore better parameterization beyond 1.5-order closure.

7) Line 5: l here is a mixing length scale so please define it as such other "length scale" is to generic.

We have revised the expression for length scale in the abstract to avoid confusion.

*Examples include multi-layer urban canopy models (MLUCMs), where the vertical variability of turbulent fluxes is calculated by solving prognostic momentum and turbulent kinetic energy (TKE, k) using mixing length scale (l) and drag parameterizations.*

The turbulent length scale is introduced from Sec. 3.1 at L155 instead.

8) Line 32: Usually when one refers to a scheme like the 1.5 order turbulence closure model, the original reference or a textbook is cited. Here the authors cite (Bougeault and Lacarrere, 1989); is that because the MLUCM uses a specific form formulation of the 1.5 order closure that was proposed in (Bougeault and Lacarrere, 1989) ?

The *k-l* formulation in the original multi-layer model from (Martilli et al. 2002) was built based on (Bougeault and Lacarrere, 1989), which parameterized orography-induced turbulence. Specifically, the formulation of turbulent and dissipation length scales from MLUCM follows (Bougeault and Lacarrere, 1989).

9) Line 44: remove "optimally"; this would usually imply a formal optimization, which LES does not do, to balance accuracy and cost. In fact, most LES go for the highest possible resolution so they pay the highest cost they can afford, so that is not an optimization.

Thanks for the comments. Indeed, from our experience the cost of LES is high. However, this research is only possible using LES because it involves triple correlation. Therefore, we changed the expression as follows,

*The most common way to enable UCMs to capture these factors is by relating the variation of urban geometric parameters to the corresponding flow statistics based on Computational Fluid Dynamics (CFD) models, **where the Large-Eddy Simulation (LES) offers a feasible way to obtain the higher moment of flow statistics.***

10) Line 263 and elsewhere: again avoid using the word optimum. This seems to be an empirical selection, and that is perfectly fine, but it is not the outcome of an optimization.

Thanks to the reviewer for the comments, there were two "optimum" expressions in the manuscript and they have been removed.

11) Line 66: some of the citation formats should be corrected. For example, here since the citation is part of the text only the year should be in parentheses.

Thanks to the reviewer for checking. We have gone through the citations to make sure the format is corrected.

12) Lines 114 and 116: the authors seem to give a Dirichelet BC on line 114 (value of s) and then a Neumann one on line 116 (flux or gradient of s). Both cannot be imposed at the same time. If I understand correctly, the one on line 116 is the actual one and line 114 is just the surface value of the initial profile but is not imposed. Please clarify.

Thanks to the reviewer for checking. Yes, L114 explains how the initial profile was prepared and the actual forcing is a surface emission flux. We have made it clear in the manuscript.

*Apart from solving momentum and pressure fields, a passive scalar was introduced with an **initial** surface value $c_0$=0.06, and a constant negative gradient of -5e-5/m **to form the initial profile.***

13) Line 51 and Eq. 7: there should be a minus sign in all these flux models for the flux to be downgradient.

Thanks to the reviewer for checking, we have added minus signs in Eq.7 and other inline equations.

14) Line 159: I think it should be : "The fifth term represents the source of TKE generated..", right? Maybe then just say D_K to make it clearer.

Thanks to the reviewer for checking, we have changed the expression to use D_K to refer to the fifth term.

15) Line 178-179: "due to the resistance difference to the constant pressure gradient between the free atmosphere and urban canopy." This statement is very confusing and unclear.

The text describes that the turbulent flux of TKE and momentum are both negative within the canopy but demonstrate differences due to different mechanisms in production and destruction. The expression has been modified as follows,

*Within the canopy, the turbulent fluxes of momentum and TKE are predominantly negative,* **reflecting downgradient transports driven by their differing production and destruction mechanisms.**

16) Line 180: Not sure what the authors mean by "Being first flow moments, the eddy diffusivity…". The eddy diffusivity is not a statistical moment of the flow field so not sure what is meant here.

We were trying to say that the scalar and momentum field is the first flow moment and they share a similar shape in the eddy-diffusivity. We have removed the expression "being the first flow moment".

*The eddy diffusivity of scalar ($K_c$) and momentum ($K_n$) maintains a similar shape as a result of the relatively simpler mechanism in their production, destruction, and transport which justifies the simplification ($K_c \sim K_m$)*

17) Caption of Fig 4, the authors use the term "dispersive velocity" but they did not formally define these. Please do.

Thanks to the reviewer for carefully checking. We have added the definition at L131:

*Then, flow properties are spatially averaged to match a grid cell of a mesoscale model (horizontal averaging, $\overline{\phi} = \overline{\langle\phi\rangle} + \widetilde{\phi}$, **where $\widetilde{\phi}$ is termed as a dispersive component**),* and in the caption of Fig. 4: *Vectors in the bottom left (c) panel show the mean velocity direction, whereas in the bottom right, they show the direction of the dispersive velocity (, $\widetilde{\phi} = \overline{\phi} - \overline{\langle\phi\rangle}$ ).*

18) Figure 5 is hard to understand. Is this the PDF rather than the CDF? The integral of the CDF over this plot should be 1, right, so not sure what these contours are. Does it mean you get a CDF of 1 for example if you integrate outside of the contour of CDF=100%?

Thanks to the reviewer for checking, yes, it's PDF rather than CDF. We were trying to do a similar plot to Fig. 5 in (Sušelj et al. 2012, https://doi.org/10.1175/JAS-D-11-090.1) but the integration starts from outside to sample the strongest fluxes. We have made this clear in the caption:

*Contour plot showing the two-dimensional histograms of the PDF of the dispersive velocities at 2.5, 5, 7.5, 10, 15, and 22 m. The integral of the PDF **(from outside to the mean value)** within the area is delineated by isolines on the logarithmic scale.*

**Response to Reviewer #2**

This manuscript provides a description of improvements to the turbulent-kinetic-energy and dispersive momentum flux terms in the urban canopy parameterisation MLUCM based on LES data. This is an important contribution to urban modelling and the manuscript is overall well written. I have some general comments and specific suggestions on clarifications:

**Response:** We thank the reviewer for their feedback on the manuscript. Please see below for responses to comments.

I find that there is a bit of a mismatch between what is discussed in the introduction (impact of complex urban geometry) and what is described in section 2 (staggered cubes with different heights). I suggest to better highlight that this study focuses on height differences as the critical variation, not on building geometry or street alignment.

Thanks to the reviewer for the suggestion. Here we are discussing the building geometry complexity to introduce the dispersive momentum flux in urban applications and aren't discussing further how to resolve the street alignment

We have modified the introduction (the paragraph starting from L64) to reflect the research focus as follows,

*Arising from spatially-averaged flow properties (that is core to MLUCM development), dispersive fluxes illustrate the transport of variables by time mean structures smaller than the averaging grid size, constituting another unique urban phenomenon (Poggi and Katul, 2008b). The strength of the dispersive flux is highly related to the horizontal (e.g., (Lu et al., 2023b)) and vertical (e.g., (Xie et al., 2008)) urban structures and exhibits substantial spatial variability (Harman et al., 2016) that forbids generalized characterizations.*

I would also like to see a comment on what the authors expect to be the impact of non-cubic buildings on their parameterisations. Maybe a comparison to the work of Blunn et al., 2022 or other relevant studies (if there are any) could be useful.

We thank the reviewer for the suggestion. We are not only considering cubes but cuboids and most buildings are a combination of cuboids. There are plenty of studies focused on non-cubic-shaped roughness beyond urban applications such as cylinders (Agbaglah and Mavriplis, 2019), and spheres (Ye et al., 2016) that exhibit different transport behavior. However, it requires a great effort to synthesize them together in a comparable manner, and non-cubic roughness representation for urban parameterization it's out of scope. We have noted this in the manuscript in Sect 3.1:

*The roughness flow over other building shapes such as cuboids (Blunn et al., 2022;Li and Bou-Zeid, 2019), cylinders (Agbaglah and Mavriplis, 2019), and spheres (Ye et al., 2016) exhibits different transport behavior. However, most buildings are a combination of cuboids (Oke et al. 2017), and non-urban roughness flow is not suitable to inform urban parameterization. Therefore, the transport behavior over non-cuboid roughness is outside of the current scope.*

Section 3 is not very clear on which terms (equations) refer to the LES model, mesoscale model or parameterisation. In particular the first part, and I don't understand the last sentence in Section 3 on what MLUCM does (line 145). Please be more specific.

Thanks to the reviewer for the notes. Section 3 aims to introduce general momentum budget equations after spatial-temporal averaging and neglecting Coriolis and buoyancy forces over the N-S equation. Therefore, Eq. 1 is not directly connected to LES or mesoscale models but is an intermediate formulation for parameterization that includes all terms to be parameterized in Sect 3.1-3.3

We have revised the manuscript to be more specific:

*The multi-layer model is designed to relate terms in Eq. 1 to the variation of urban surface, where the vertical diffusion of momentum (Sect. 3.1 and Sect. 3.2) and drag (Sect. 3.3) are parameterized.*

Please ensure all your figures have subfigure labels which are referenced in the main text, and the notation used is properly described either in the caption or main text.

Thanks to the reviewer for the suggestion. We have checked figure labels and notations to ensure consistency.

Abstract, line 11: […] we conducted 49 large-eddy simulations … 'conducted' suggest that you present the results here for the first time, but they have been presented in another paper, correct? Please be clear on what is the contribution of this manuscript. I suggest to use 'analysed'.

Thanks to the reviewer for the suggestion. We have changed the expression and elsewhere to make sure it's not misleading.

Line 16: MLUCM v2.0 has not been introduced at this point, please add a sentence where you introduce this particular model.

Thanks to the reviewer for the suggestion. Indeed, the V2.0 model is not fully introduced at that point. We have changed the expression to avoid confusion.

*In response to these findings, we propose two changes* **to the previous version of** *MLUCM: (a) separate characterization for turbulent diffusion coefficient for momentum and TKE;*

Introduction, line 76: Related to above. I find it a bit off to talk about the multi-layer model when referring to a specific model, since there are different urban multi-layer models.

Thanks to the reviewer for the suggestion, indeed, there are other multi-layer urban canopy models too. We have revised the manuscript to reflect that and then go to the specific model as follows:

***Multi-layer UCMs are fundamentally more versatile than single-layer UCMs in characterizing the urban effects (Hendricks et al., 2020) and in-canopy processes and have evolved into different implementations (Kondo et al., 2005; Yuan et al., 2019;Santiago and Martilli, 2010b). The Building Effects Parameterization (BEP, (Martilli et al., 2002)) model and developments to its flow parameterization through the multi-layer urban canopy model (MLUCM, (Nazarian et al., 2020)) have been directed***

*toward multiple applications such as simulation of building-tree interaction (BEP-tree, Krayenhoff et al. (2020)) interaction of indoor building energy exchanges with outdoor climate (BEP-BEM, (Salamanca et al., 2010)).*

Section 3, line 128: "a common approach in this situation…" This sentence is confusing, what situation do you mean? Do you mean a common approach to derive an urban parameterization?

Thanks to the reviewer for checking. We have revised the expression as follows:

*A common approach **to derive parameterizations** is to apply a time-averaging*

Line 133: the notation of the spatial averaging is not clear, please state what the two decomposed terms on the right-hand side are.

Thanks to the reviewer for the suggestion. We have revised the averaging expression for clarification.

*For momentum, Reynolds decomposition is first applied to the 3-dimensional instantaneous equations that decompose mean flow quantities ($\overline{\phi}$) from their fluctuating components ($\phi'$, time or ensemble averaging, $\phi = \overline{\phi} + \phi'$ ). Then, flow properties are spatially averaged to match a grid cell of a mesoscale model (horizontal averaging, $\overline{\phi} = \widetilde{\phi} + \overline{\langle\phi\rangle}$ , where $\widetilde{\phi}$ is termed as a dispersive component).*

Lines 133-136: I find the justification for using the intrinsic average not very clear. Personally, I think the comprehensive average is more suitable for vertical parameterisations, since it keeps the grid-box volume constant across height and just describes the change of the parameters with height (the buildings are never "physically" in the model). Using the intrinsic average gives you these strange-looking kinks at the canopy top in Fig. 1 and Fig.9. But this is just a comment, I'm not suggesting changing it.

Thanks to the reviewer for the suggestion. Indeed, using intrinsic averaging, we encountered some twists on the profiles. There are no specific reasons for this choice, but to keep consistency between model versions in deriving the parameterization

Line 141: "… represents vertical transport events"

Thanks for checking, we have revised the manuscript as follows,

*The first term on the RHS of both equations represents **vertical** transport events.*

Section 3.1, line 197: What do you mean by "that do not break the paramterization in the present study"?

Thanks to the reviewer for checking. I was trying to say the inversion for high-density cases may not exist in a realistic neighborhood.

*However, realistic neighborhoods with extremely high density typically exhibit varied horizontal arrangements, leading to distinct flow characteristics compared to idealized building blocks (Lu et al., 2023b).*

Figure 1: epsilon is not defined. Use a, b, c for 1st, 2nd, 3rd column. Would be even clearer to label each subfigure with a--i and refer to the individual subfigures in the main text.

Thanks to the reviewer for checking, the epsilon should be gamma, as explained in Eq 7. We have added a, b, and c for 1st, 2nd, and 3rd columns for Fig.1

Line 205: "(note the vertical range of two rows in Fig 1)" I don't understand what this means. Consider using labels for each subfigure and refer to them specifically.

Thanks to the reviewer for checking. It should be Figure 2, and we asked the audience to pay attention to the difference in vertical ranges of momentum and TKE. We have changed the expression as follows and added labels to subfigure in Figure 2 and elsewhere applicable.

*The magnitude of eddy-diffusivity for TKE is ~ 3.5 times higher than that for momentum (note the different vertical ranges **for momentum and TKE** in Fig. 2).*

Paragraph from Line 204: If you are referring to what is shown in Figure 2 here, I believe it would really help to have labels for each subfigure and refer to them explicitly in this paragraph.

Thanks to the reviewer for checking. We have added labels to subfigures and updated the discussion correspondingly.

Section 3.2, line 247 (equation 8): This equation needs to be embedded in a sentence, e.g.: "The dispersive flux of momentum then reads: …"

We thank the reviewer for the careful check. Eq 8 is mentioned before Eq 10, but it's better to make it clear the connection between derivations above Eq. 8. We have added a sentence:

***The dispersive flux of momentum then reads:***

Figure 4: Caption contains labels a--d (good!) but the figure does not.

Thanks to the reviewer for checking. We have added labels in Figure 4 and checked elsewhere for similar issues.

Paragraph from line 254: I don't think the sampling approach has been explained here, and what do the cut-off values do? Maybe I missed it.

The sampling is fulfilled by adjusting the CDF value from 1 (the entire field) to 0 as the dispersive flux is only strong near the building facets and has to be sampled out from the 2D field. Here are the detailed steps:
1. Separate downward and upward fluxes as two lists;
2. Sort these two lists into a descending order in terms of the magnitude;
3. Do integration from the beginning, so CDF 0—>1;

4. As the CDF is growing, check the enclosed region (larger as the CDF goes larger) until it encloses the windward and leeward structure to find the cut-off value.
5. The cut-off value indicates the position in these two lists where we stop recognizing the flux as directly induced by buildings.
6. Do the averaging based on the cut-off value, as plotted in Fig. 6
7. Find appropriate scaling parameters and do parameterization.

We have integrated these steps into the appendix as follows,

*The sampling procedure to extract the components of building-induced dispersive fluxes was conducted as follows: a) Separating the dispersive fluxes into gradient (downward) and counter-gradient (upward) components; b) Sorting these two fluxes into a descending order in terms of the magnitude; c) lowering the cumulative density function of the two fluxes from the largest allows a smaller subdomain to be sampled until subdomains only enclose transport events connected to buildings; d) Spatially averaging the sampled dispersive fluxes over subdomains for both gradient and counter-gradient dispersive momentum flux.*

Section 3.3, line 316: 'mounted' sounds odd.

Thanks, we have removed the word to avoid confusion.

Section 4, Table 2: What do you mean by "to Km" under dispersive momentum flux?

"To Km" means the dispersive flux was considered as an increment to the turbulent flux in the evaluation of Km. We have revised the caption in the table to make it clear:

*Therefore, the 1D-Original does not differentiate turbulent TKE flux from turbulent momentum flux and lumps the dispersive momentum flux into the calculation of $K_m$ (To $K_m$).*

Line 366-368: I would add a sentence saying that you quantify this improvement in the paragraphs below by looking at the RMSE between the LES data and the models.

We thank the reviewer for the suggestion. Now there is a connecting sentence to introduce RMSE as an indicator of performance.

*To evaluate the comprehensive performance of scenarios over different height configurations, Fig. 10 presents the root mean square error (RMSE) between vertical profiles (ranging from z=[0-3H$_{mean}$]) of LES and four MLUCM scenarios averaged over seven configurations (one uniform and six variable height, as shown in Table 1) over seven densities.*

Line 370; Line 380-382: It is hard to spot the difference in velocity profiles. Where is the improvement coming from, i.e. at which height levels? Can you say what might be the reason for it?

The RMSE is evaluated from the ground to 3Hmean, which represents the range as shown in Fig. 9 showing the vertical profile comparison.

For example, the difference for S3 (purple lines) came from the overall higher velocity which is more obvious in the dense layout. S3 has higher velocity because the newly introduced MF term is an explicit source of momentum, while differentiation of Kk from Km also serves to enhance the momentum transport from the canopy interface to the canopy. Therefore, both measures collectively lead to higher streamwise velocity. We have revised the manuscript to make the interpretation easier:

**The prediction on streamwise velocity does not present a great variation where 1D-$K_k$-EDMF exhibits the highest value. This is explainable by the newly introduced MF term as an explicit source of momentum and differentiation of $K_k$ from $K_m$ enhancing the momentum transport from the atmosphere to the canopy.**

Line 371: I suggest spelling out 'MF'.

Thanks, we have added the complete name for MF (mass-flux).

Line 372: The dispersive flux is not (only) a source of momentum, but also a sink term.

We thank the reviewer for the suggestion. Indeed, dispersive fluxes act as a source/sink of momentum at different vertical levels and for different layouts. However, the parameterized component of dispersive momentum flux as an MF term is a source of momentum. We have revised the manuscript to make it clear:

*As a result, **1D-$K_k$-EDMF** yields an even higher TKE due to the **introduction of an MF term to parameterize dispersive momentum flux as an explicit source leading to higher TKE production.***

Line 385: Can you speculate on why the turbulent fluxes behave contrary to the TKE (i.e., better in sparse layouts)?

The turbulent momentum fluxes are diagnostically obtained in MLUCM from the *k-l* closure.

$$\langle \overline{u'w'} \rangle = -C_k l_k \langle \overline{k} \rangle^{1/2} \frac{\partial \langle \overline{u} \rangle}{\partial z},$$

We understand the reviewer's concern that better TKE estimation should lead to better momentum fluxes estimation. There are competing factors in MLUCM in controlling the performance:

Zooming in on the Tuw profile within the canopy, the **S2** and **S3** scenario has a lower magnitude than the **original**, which indicates the vertical gradient of streamwise velocity is smaller. This can be explained by S2 and S3 enhancing the momentum exchange by introducing EDMF and Kk. As a result, the momentum will be more efficiently mixed and has a steeper profile that results in smaller vertical gradients.

We have made this clear in the manuscript:

*The turbulent momentum flux is diagnostically obtained in MLUCM based on the k-l closure $\langle \overline{u'w'} \rangle = -C_k l_k \langle \overline{k} \rangle^{1/2} \frac{\partial \langle \overline{u} \rangle}{\partial z}$ that is highly related to streamwise velocity and TKE. However, it exhibits a consistently contrasting performance trend compared to TKE which leads to less improvement in denser layouts. This pattern is explainable by both improvements (Table 2) enhancing the vertical exchange of flow properties within the canopy, which results in a smaller vertical gradient of streamwise velocity.*

Figure 10: I suggest showing this Figure as the difference to the original scheme (i.e., S1 – original, S2 – original, …) and on the y-axis as percentage of improvement. In this bar chart it is hard to identify the improvements and their significance.

We thank the reviewer for the suggestion. Figure 10 has been revised to show RMSE difference between the original and S1-S3 (RMSE_**original** - RMSE_**refinement**, so positive RMSE means better performance from **refinement**), and the corresponding description in the manuscript has been revised too.

[Figure]

Section 5, line 388: "This [study] refined the characterization …"

Thanks, we have added the word.

Appendix: Figure A1 needs a better caption, explaining what the yellow block is, what the cut-off values are.

Thanks to the reviewer for checking, the color for blocks shows the building height. We have added a separate color bar for this and elaborated on the details in the caption.

*Sampling regions of dispersive fluxes for five co∈[0.3,0.4,0.5,0.6,0.7]) cut-off CDF values **as discussed in Sect. 3.2. Higher cut-off values for CDF lead to broader sampling regions and lower mean strength in the mass-flux term.***